# A Stochastic Derivative Free Optimization Method with Momentum

**Eduard Gorbunov**[*]
MIPT, Russia and IITP RAS, Russia and RANEPA, Russia
`eduard.gorbunov@phystech.edu`

**Adel Bibi**
KAUST, Saudi Arabia
`adel.bibi@kaust.edu.sa`

**Ozan Sener**
Intel Labs
`ozan.sener@intel.com`

**El Houcine Bergou**
KAUST, Saudi Arabia and MaIAGE, INRA, France
`elhoucine.bergou@inra.fr`

**Peter Richtárik**
KAUST, Saudi Arabia and MIPT, Russia
`peter.richtarik@kaust.edu.sa`

## Abstract

We consider the problem of unconstrained minimization of a smooth objective function in $\mathbb{R}^d$ in setting where only function evaluations are possible. We propose and analyze stochastic zeroth-order method with heavy ball momentum. In particular, we propose, `SMTP`, a momentum version of the stochastic three-point method (`STP`) Bergou et al. (2019). We show new complexity results for non-convex, convex and strongly convex functions. We test our method on a collection of learning to continuous control tasks on several MuJoCo Todorov et al. (2012) environments with varying difficulty and compare against `STP`, other state-of-the-art derivative-free optimization algorithms and against policy gradient methods. `SMTP` significantly outperforms `STP` and all other methods that we considered in our numerical experiments. Our second contribution is `SMTP` with importance sampling which we call `SMTP_IS`. We provide convergence analysis of this method for non-convex, convex and strongly convex objectives.

## 1 Introduction

In this paper, we consider the following minimization problem

$$\min_{x \in \mathbb{R}^d} f(x), \tag{1}$$

where $f : \mathbb{R}^d \to \mathbb{R}$ is "smooth" but not necessarily a convex function in a Derivative-Free Optimization (DFO) setting where only function evaluations are possible. The function $f$ is bounded from below by $f(x^*)$ where $x^*$ is a minimizer. Lastly and throughout the paper, we assume that $f$ is $L$-smooth.

**DFO.** In DFO setting Conn et al. (2009); Kolda et al. (2003), the derivatives of the objective function $f$ are not accessible. That is they are either impractical to evaluate, noisy (function $f$ is noisy) (Chen, 2015) or they are simply not available at all. In standard applications of DFO, evaluations of $f$ are only accessible through simulations of black-box engine or software as in reinforcement learning and continuous control environments Todorov et al. (2012). This setting of optimization problems appears also in applications from computational medicine Marsden et al. (2008) and fluid dynamics Allaire (2001); Haslinger & Mäckinen (2003); Mohammadi & Pironneau (2001) to localization Marsden et al. (2004; 2007) and continuous control Mania et al. (2018); Salimans et al. (2017) to name a few.

The literature on DFO for solving (1) is long and rich. The first approaches were based on deterministic direct search (DDS) and they span half a century of work Hooke & Jeeves (1961); Su (1979); Torczon

---

[*]The research of Eduard Gorbunov was supported by RFBR, project number 18-31-20005 mol_a_ved

(1997). However, for DDS methods complexity bounds have only been established recently by the work of Vicente and coauthors Vicente (2013); Dodangeh & Vicente (2016). In particular, the work of Vicente Vicente (2013) showed the first complexity results on non-convex $f$ and the results were extended to better complexities when $f$ is convex Dodangeh & Vicente (2016). However, there have been several variants of DDS, including randomized approaches Matyas (1965); Karmanov (1974a;b); Baba (1981); Dorea (1983); Sarma (1990). Only very recently, complexity bounds have also been derived for randomized methods Diniz-Ehrhardt et al. (2008); Stich et al. (2011); Ghadimi & Lan (2013); Ghadimi et al. (2016); Gratton et al. (2015). For instance, the work of Diniz-Ehrhardt et al. (2008); Gratton et al. (2015) imposes a decrease condition on whether to accept or reject a step of a set of random directions. Moreover, Nesterov & Spokoiny (2017) derived new complexity bounds when the random directions are normally distributed vectors for both smooth and non-smooth $f$. They proposed both accelerated and non-accelerated zero-order (ZO) methods. Accelerated derivative-free methods in the case of inexact oracle information was proposed in Dvurechensky et al. (2017). An extension of Nesterov & Spokoiny (2017) for non-Euclidean proximal setup was proposed by Gorbunov et al. (2018) for the smooth stochastic convex optimization with inexact oracle. In Stich (2014a;b) authors also consider acceleration of ZO methods and, in particular, develop the method called SARP, proved that its convergence rate is not worse than for non-accelerated ZO methods and showed that in some cases it works even better.

More recently and closely related to our work, Bergou et al. (2019) proposed a new randomized direct search method called *Stochastic Three Points* (STP). At each iteration $k$ STP generates a random search direction $s_k$ according to a certain probability law and compares the objective function at three points: current iterate $x_k$, a point in the direction of $s_k$ and a point in the direction of $-s_k$ with a certain step size $\alpha_k$. The method then chooses the best of these three points as the new iterate:

$$x_{k+1} = \arg\min\{f(x_k), f(x_k + \alpha_k s_k), f(x_k - \alpha_k s_k)\}.$$

The key properties of STP are its simplicity, generality and practicality. Indeed, the update rule for STP makes it extremely simple to implement, the proofs of convergence results for STP are short and clear and assumptions on random search directions cover a lot of strategies of choosing decent direction and even some of first-order methods fit the STP scheme which makes it a very flexible in comparison with other zeroth-order methods (e.g. two-point evaluations methods like in Nesterov & Spokoiny (2017), Ghadimi & Lan (2013), Ghadimi et al. (2016), Gorbunov et al. (2018) that try to approximate directional derivatives along random direction at each iteration). Motivated by these properties of STP we focus on further developing of this method.

**Momentum.** Heavy ball momentum[1] is a special technique introduced by Polyak in 1964 Polyak (1964) to get faster convergence to the optimum for the first-order methods. In the original paper, Polyak proved that his method converges *locally* with $O\left(\sqrt{L/\mu}\log 1/\varepsilon\right)$ rate for twice continuously differentiable $\mu$-strongly convex and $L$-smooth functions. Despite the long history of this approach, there is still an open question whether heavy ball method converges to the optimum *globally* with accelerated rate when the objective function is twice continuous differentiable, $L$-smooth and $\mu$-strongly convex. For this class of functions, only non-accelerated global convergence was proved Ghadimi et al. (2015) and for the special case of quadratic strongly convex and $L$-smooth functions Lessard et. al. Lessard et al. (2016) recently proved asymptotic accelerated global convergence. However, heavy ball method performs well in practice and, therefore, is widely used. One can find more detailed survey of the literature about heavy ball momentum in Loizou & Richtárik (2017).

**Importance Sampling.** Importance sampling has been celebrated and extensively studied in stochastic gradient based methods Zhao & Zhang (2015) or in coordinate based methods Richtárik & Takáč (2016). Only very recently, Bibi et al. (2019) proposed, STP_IS, the first DFO algorithm with importance sampling. In particular, under coordinate-wise smooth function, they show that sampling coordinate directions, can be generalized to arbitrary directions, with probabilities proportional to the function coordinate smoothness constants, improves the leading constant by the same factor typically gained in gradient based methods.

**Contributions.** Our contributions can be summarized into three folds.

- **First ZO method with heavy ball momentum.** Motivated by practical effectiveness of first-order momentum heavy ball method, we introduce momentum into STP method and

---

[1]We will refer to this as momentum.

---

**Algorithm 1** SMTP: Stochastic Momentum Three Points

---

**Require:** learning rates $\{\gamma^k\}_{k\geq 0}$, starting point $x^0 \in \mathbb{R}^d$, $\mathcal{D}$ — distribution on $\mathbb{R}^d$, $0 \leq \beta < 1$ — momentum parameter
1: Set $v^{-1} = 0$ and $z^0 = x^0$
2: **for** $k = 0, 1, \ldots$ **do**
3:     Sample $s^k \sim \mathcal{D}$
4:     Let $v_+^k = \beta v^{k-1} + s^k$ and $v_-^k = \beta v^{k-1} - s^k$
5:     Let $x_+^{k+1} = x^k - \gamma^k v_+^k$ and $x_-^{k+1} = x^k - \gamma^k v_-^k$
6:     Let $z_+^{k+1} = x_+^{k+1} - \frac{\gamma^k \beta}{1-\beta} v_+^k$ and $z_-^{k+1} = x_-^{k+1} - \frac{\gamma^k \beta}{1-\beta} v_-^k$
7:     Set $z^{k+1} = \arg\min \left\{ f(z^k), f(z_+^{k+1}), f(z_-^{k+1}) \right\}$
8:     Set $x^{k+1} = \begin{cases} x_+^{k+1}, & \text{if } z^{k+1} = z_+^{k+1} \\ x_-^{k+1}, & \text{if } z^{k+1} = z_-^{k+1} \\ x^k, & \text{if } z^{k+1} = z^k \end{cases}$ and $v^{k+1} = \begin{cases} v_+^{k+1}, & \text{if } z^{k+1} = z_+^{k+1} \\ v_-^{k+1}, & \text{if } z^{k+1} = z_-^{k+1} \\ v^k, & \text{if } z^{k+1} = z^k \end{cases}$
9: **end for**

---

propose new DFO algorithm with heavy ball momentum (SMTP). We summarized the method in Algorithm 1, with theoretical guarantees for non-convex, convex and strongly convex functions under generic sampling directions $\mathcal{D}$. We emphasize that the SMTP with momentum is not a straightforward generalization of STP and Polyak's method and requires insights from virtual iterates analysis from Yang et al. (2016).

To the best of our knowledge it is the first analysis of derivative-free method with heavy ball momentum, i.e. we show that the same momentum trick that works for the first order method could be applied for zeroth-order methods as well.

- **First ZO method with both heavy ball momentum and importance sampling.** In order to get more gain from momentum in the case when the sampling directions are coordinate directions and the objective function is coordinate-wise $L$-smooth (see Assumption 4.1), we consider importance sampling to the above method. In fact, we propose the first zeroth-order momentum method with importance sampling (SMTP_IS) summarized in Algorithm 2 with theoretical guarantees for non-convex, convex and strongly convex functions. The details and proofs are left for Section 4 and Appendix E.

- **Practicality.** We conduct extensive experiments on continuous control tasks from the MuJoCo suite Todorov et al. (2012) following recent success of DFO compared to model-free reinforcement learning Mania et al. (2018); Salimans et al. (2017). We achieve with SMTP_IS the state-of-the-art results on across all tested environments on the continuous control outperforming DFO Mania et al. (2018) and policy gradient methods Schulman et al. (2015); Rajeswaran et al. (2017).

We provide more detailed comparison of SMTP and SMTP_IS in Section E.4 of the Appendix.

## 2 NOTATION AND DEFINITIONS

We use $\| \cdot \|_p$ to define $\ell_p$-norm of the vector $x \in \mathbb{R}^d$: $\|x\|_p \stackrel{\text{def}}{=} \left( \sum_{i=1}^d |x_i|^p \right)^{1/p}$ for $p \geq 1$ and $\|x\|_\infty \stackrel{\text{def}}{=} \max_{i \in [d]} |x_i|$ where $x_i$ is the $i$-th component of vector $x$, $[d] = \{1, 2, \ldots, d\}$. Operator $\mathbf{E}[\cdot]$ denotes mathematical expectation with respect to all randomness and $\mathbf{E}_{s \sim \mathcal{D}}[\cdot]$ denotes conditional expectation w.r.t. randomness coming from random vector $s$ which is sampled from probability distribution $\mathcal{D}$ on $\mathbb{R}^d$. To denote standard inner product of two vectors $x, y \in \mathbb{R}^d$ we use $\langle x, y \rangle \stackrel{\text{def}}{=} \sum_{i=1}^d x_i y_i$, $e_i$ denotes $i$-th coordinate vector from standard basis in $\mathbb{R}^d$, i.e. $x = \sum_{i=1}^d x_i e_i$. We use $\| \cdot \|^*$ to define the conjugate norm for the norm $\| \cdot \|$: $\|x\|^* \stackrel{\text{def}}{=} \max \left\{ \langle a, x \rangle \mid a \in \mathbb{R}^d, \|a\| \leq 1 \right\}$.

As we mention in the introduction we assume throughout the paper[2] that the objective function $f$ is $L$-smooth.

---

[2] We will use thinner assumption in Section 4.

| Assumptions on $f$ | SMTP Complexity | Theorem | Importance Sampling | SMTP_IS Complexity | Theorem |
|---|---|---|---|---|---|
| None | $\frac{2r_0 L \gamma_{\mathcal{D}}}{\mu_{\mathcal{D}}^2 \varepsilon^2}$ | 3.1 | $p_i = \frac{L_i}{\sum_{i=1}^d L_i}$ | $\frac{2r_0 d \sum_{i=1}^d L_i}{\varepsilon^2}$ | E.1 |
| Convex, $R_0 < \infty$ | $\frac{1}{\varepsilon} \frac{L \gamma_{\mathcal{D}} R_0^2}{\mu_{\mathcal{D}}^2} \ln\left(\frac{2r_0}{\varepsilon}\right)$ | 3.2 | $p_i = \frac{L_i}{\sum_{i=1}^d L_i}$ | $\frac{R_0^2 d \sum_{i=1}^d L_i}{\varepsilon} \ln\left(\frac{2r_0}{\varepsilon}\right)$ | E.2 |
| $\mu$-strongly convex | $\frac{L}{\mu \mu_{\mathcal{D}}^2} \ln\left(\frac{2r_0}{\varepsilon}\right)$ | 3.5 | $p_i = \frac{L_i}{\sum_{i=1}^d L_i}$ | $\frac{\sum_{i=1}^d L_i}{\mu} \ln\left(\frac{2r_0}{\varepsilon}\right)$ | E.5 |

Table 1: Summary of the new derived complexity results of SMTP and SMTP_IS. The complexities for SMTP are under a generic sampling distribution $\mathcal{D}$ satisfying Assumption 3.1 while for SMTP_IS are under an arbitrary discrete sampling from a set of coordinate directions following Bibi et al. (2019) where we propose an importance sampling that improves the leading constant marked in red. Note that $r_0 = f(x_0) - f(x_*)$ and that all assumptions listed are in addition to Assumption 2.1. Complexity means number of iterations in order to guarantee $\mathbf{E}\|\nabla f(\bar{z}^K)\|_{\mathcal{D}} \leq \varepsilon$ for the non-convex case, $\mathbf{E}\left[f(z^K) - f(x^*)\right] \leq \varepsilon$ for convex and strongly convex cases. $R_0 < \infty$ is the radius in $\|\cdot\|_{\mathcal{D}}^*$-norm of a bounded level set where the exact definition is given in Assumption 3.2. We notice that for SMTP_IS $\|\cdot\|_{\mathcal{D}} = \|\cdot\|_1$ and $\|\cdot\|_{\mathcal{D}}^* = \|\cdot\|_\infty$ in non-convex and convex cases and $\|\cdot\|_{\mathcal{D}} = \|\cdot\|_2$ in the strongly convex case.

**Assumption 2.1.** *(L-smoothness) We say that $f$ is L-smooth if*

$$\|\nabla f(x) - \nabla f(y)\|_2 \leq L\|x - y\|_2 \quad \forall x, y \in \mathbb{R}^d. \tag{2}$$

From this definition one can obtain

$$|f(y) - f(x) - \langle \nabla f(x), y - x \rangle| \leq \frac{L}{2}\|y - x\|_2^2, \quad \forall x, y \in \mathbb{R}^d, \tag{3}$$

and if additionally $f$ is convex, i.e. $f(y) \geq f(x) + \langle \nabla f(x), y - x \rangle$, we have

$$\|\nabla f(x)\|_2^2 \leq 2L(f(x) - f(x^*)), \quad \forall x \in \mathbb{R}^d. \tag{4}$$

# 3 STOCHASTIC MOMENTUM THREE POINTS (SMTP)

Our analysis of SMTP is based on the following key assumption.

**Assumption 3.1.** *The probability distribution $\mathcal{D}$ on $\mathbb{R}^d$ satisfies the following properties:*

1. *The quantity $\gamma_{\mathcal{D}} \stackrel{def}{=} \mathbf{E}_{s \sim \mathcal{D}}\|s\|_2^2$ is finite.*

2. *There is a constant $\mu_{\mathcal{D}} > 0$ for a norm $\|\cdot\|_{\mathcal{D}}$ in $\mathbb{R}^d$ such that for all $g \in \mathbb{R}^d$*

$$\mathbf{E}_{s \sim \mathcal{D}}|\langle g, s \rangle| \geq \mu_{\mathcal{D}}\|g\|_{\mathcal{D}}. \tag{5}$$

Some examples of distributions that meet above assumption are described in Lemma 3.4 from Bergou et al. (2019). For convenience we provide the statement of the lemma in the Appendix (see Lemma F.1).

Recall that one possible view on STP Bergou et al. (2019) is as following. If we substitute gradient $\nabla f(x^k)$ in the update rule for the gradient descent $x^{k+1} = x^k - \gamma^k \nabla f(x^k)$ by $\pm s^k$ where $s^k$ is sampled from distribution $\mathcal{D}$ satisfied Assumption 3.1 and then select $x^{k+1}$ as the best point in terms of functional value among $x^k, x^k - \gamma^k s^k, x^k + \gamma^k s^k$ we will get exactly STP method. However, gradient descent is not the best algorithm to solve unconstrained smooth minimization problems and the natural idea is to try to perform the same substitution-trick with more efficient first-order methods than gradient descent.

We put our attention on Polyak's heavy ball method where the update rule could be written in the following form:

$$v^k = \beta v^{k-1} + \nabla f(x^k), \quad x^{k+1} = x^k - \gamma^k v^k. \tag{6}$$

As in STP, we substitute $\nabla f(x^k)$ by $\pm s^k$ and consider new sequences $\{v_+^k\}_{k \geq 0}$ and $\{v_-^k\}_{k \geq 0}$ defined in the Algorithm 1. However, it is not straightforward how to choose next $x^{k+1}$ and $v^k$ and

the virtual iterates analysis Yang et al. (2016) hints the update rule. We consider new iterates $z_+^{k+1} = x_+^{k+1} - \frac{\gamma^k \beta}{1-\beta} v_+^k$ and $z_-^{k+1} = x_-^{k+1} - \frac{\gamma^k \beta}{1-\beta} v_-^k$ and define $z^{k+1}$ as $\arg\min \left\{ f(z^k), f(z_+^{k+1}), f(z_-^{k+1}) \right\}$. Next we update $x^{k+1}$ and $v^k$ in order to have the same relationship between $z^{k+1}, x^{k+1}$ and $v^k$ as between $z_+^{k+1}, x_+^{k+1}$ and $v_+^k$ and $z_-^{k+1}, x_-^{k+1}$ and $v_-^k$. Such scheme allows easily apply virtual iterates analysis and and generalize Key Lemma from Bergou et al. (2019) which is the main tool in the analysis of STP.

By definition of $z^{k+1}$, we get that the sequence $\{f(z^k)\}_{k \geq 0}$ is monotone:

$$f(z^{k+1}) \leq f(z^k) \qquad \forall k \geq 0. \tag{7}$$

Now, we establish the key result which will be used to prove the main complexity results and remaining theorems in this section.

**Lemma 3.1.** *Assume that $f$ is $L$-smooth and $\mathcal{D}$ satisfies Assumption 3.1. Then for the iterates of* STP *the following inequalities hold:*

$$f(z^{k+1}) \leq f(z^k) - \frac{\gamma^k}{1-\beta} |\langle \nabla f(z^k), s^k \rangle| + \frac{L(\gamma^k)^2}{2(1-\beta)^2} \|s^k\|_2^2 \tag{8}$$

*and*

$$\mathbf{E}_{s^k \sim \mathcal{D}} \left[ f(z^{k+1}) \right] \leq f(z^k) - \frac{\gamma^k \mu_{\mathcal{D}}}{1-\beta} \|\nabla f(z^k)\|_{\mathcal{D}} + \frac{L(\gamma^k)^2 \gamma_{\mathcal{D}}}{2(1-\beta)^2}. \tag{9}$$

### 3.1 NON-CONVEX CASE

In this section, we show our complexity results for Algorithm 1 in the case when f is allowed to be non-convex. In particular, we show that SMTP in Algorithm 1 guarantees complexity bounds with the same order as classical bounds, i.e. $1/\sqrt{K}$ where $K$ is the number of iterations, in the literature. We notice that query complexity (i.e. number of oracle calls) of SMTP coincides with its iteration complexity up to numerical constant factor. For clarity and completeness, proofs are left for the appendix.

**Theorem 3.1.** *Let Assumptions 2.1 and 3.1 be satisfied. Let* SMTP *with $\gamma^k \equiv \gamma > 0$ produce points $\{z^0, z^1, \ldots, z^{K-1}\}$ and $\bar{z}^K$ is chosen uniformly at random among them. Then*

$$\mathbf{E} \left[ \|\nabla f(\bar{z}^K)\|_{\mathcal{D}} \right] \leq \frac{(1-\beta)(f(x^0) - f(x^*))}{K \gamma \mu_{\mathcal{D}}} + \frac{L \gamma \gamma_{\mathcal{D}}}{2 \mu_{\mathcal{D}}(1-\beta)}. \tag{10}$$

*Moreover, if we choose $\gamma = \frac{\gamma_0}{\sqrt{K}}$ the complexity* (10) *reduces to*

$$\mathbf{E} \left[ \|\nabla f(\bar{z}^K)\|_{\mathcal{D}} \right] \leq \frac{1}{\sqrt{K}} \left( \frac{(1-\beta)(f(z^0) - f(x^*))}{\gamma_0 \mu_{\mathcal{D}}} + \frac{L \gamma_0 \gamma_{\mathcal{D}}}{2 \mu_{\mathcal{D}}(1-\beta)} \right). \tag{11}$$

*Then $\gamma_0 = \sqrt{\frac{2(1-\beta)^2(f(x^0) - f(x^*))}{L \gamma_{\mathcal{D}}}}$ minimizes the right-hand side of* (11) *and for this choice we have*

$$\mathbf{E} \left[ \|\nabla f(\bar{z}^K)\|_{\mathcal{D}} \right] \leq \frac{\sqrt{2 \left( f(x^0) - f(x^*) \right) L \gamma_{\mathcal{D}}}}{\mu_{\mathcal{D}} \sqrt{K}}. \tag{12}$$

In other words, the above theorem states that SMTP converges no worse than STP for non-convex problems to the stationary point. In the next sections we also show that theoretical convergence guarantees for SMTP are not worse than for STP for convex and strongly convex problems. However, in practice SMTP significantly outperforms STP. So, the relationship between SMTP and STP correlates with the known in the literature relationship between Polyak's heavy ball method and gradient descent.

### 3.2 CONVEX CASE

In this section, we present our complexity results for Algorithm 1 when $f$ is convex. In particular, we show that this method guarantees complexity bounds with the same order as classical bounds, i.e. $1/K$, in the literature. We will need the following additional assumption in the sequel.

**Assumption 3.2.** *We assume that $f$ is convex, has a minimizer $x^*$ and has bounded level set at $x^0$:*

$$R_0 \overset{def}{=} \max \left\{ \|x - x^*\|_{\mathcal{D}}^* \mid f(x) \leq f(x^0) \right\} < +\infty, \tag{13}$$

*where $\|\xi\|_{\mathcal{D}}^* \overset{def}{=} \max \left\{ \langle \xi, x \rangle \mid \|x\|_{\mathcal{D}} \leq 1 \right\}$ defines the dual norm to $\| \cdot \|_{\mathcal{D}}$.*

From the above assumption and Cauchy-Schwartz inequality we get the following implication:

$$f(x) \leq f(x_0) \implies f(x) - f(x_*) \leq \langle \nabla f(x), x - x^* \rangle \leq \|\nabla f(x)\|_{\mathcal{D}} \|x - x^*\|_{\mathcal{D}}^* \leq R_0 \|\nabla f(x)\|_{\mathcal{D}},$$

which implies

$$\|\nabla f(x)\|_{\mathcal{D}} \geq \frac{f(x) - f(x^*)}{R_0} \qquad \forall x : f(x) \leq f(x_0). \tag{14}$$

**Theorem 3.2** (Constant stepsize). *Let Assumptions 2.1, 3.1 and 3.2 be satisfied. If we set $\gamma^k \equiv \gamma < \frac{(1-\beta)R_0}{\mu_{\mathcal{D}}}$, then for the iterates of* SMTP *method the following inequality holds:*

$$\mathbf{E}\left[ f(z^k) - f(x^*) \right] \leq \left( 1 - \frac{\gamma \mu_{\mathcal{D}}}{(1-\beta)R_0} \right)^k \left( f(x^0) - f(x^*) \right) + \frac{L\gamma\gamma_{\mathcal{D}}R_0}{2(1-\beta)\mu_{\mathcal{D}}}. \tag{15}$$

*If we choose $\gamma = \frac{\varepsilon(1-\beta)\mu_{\mathcal{D}}}{L\gamma_{\mathcal{D}}R_0}$ for some $0 < \varepsilon \leq \frac{L\gamma_{\mathcal{D}}R_0^2}{\mu_{\mathcal{D}}^2}$ and run* SMTP *for $k = K$ iterations where*

$$K = \frac{1}{\varepsilon} \frac{L\gamma_{\mathcal{D}}R_0^2}{\mu_{\mathcal{D}}^2} \ln \left( \frac{2(f(x^0) - f(x^*))}{\varepsilon} \right), \tag{16}$$

*then we will get $\mathbf{E}\left[ f(z^K) \right] - f(x^*) \leq \varepsilon$.*

In order to get rid of factor $\ln \frac{2(f(x^0) - f(x^*))}{\varepsilon}$ in the complexity we consider decreasing stepsizes.

**Theorem 3.3** (Decreasing stepsizes). *Let Assumptions 2.1, 3.1 and 3.2 be satisfied. If we set $\gamma^k = \frac{2}{\alpha k + \theta}$, where $\alpha = \frac{\mu_{\mathcal{D}}}{(1-\beta)R_0}$ and $\theta \geq \frac{2}{\alpha}$, then for the iterates of* SMTP *method the following inequality holds:*

$$\mathbf{E}\left[ f(z^k) \right] - f(x^*) \leq \frac{1}{\eta k + 1} \max \left\{ f(x^0) - f(x^*), \frac{2L\gamma_{\mathcal{D}}}{\alpha\theta(1-\beta)^2} \right\}, \tag{17}$$

*where $\eta \overset{def}{=} \frac{\alpha}{\theta}$. Then, if we choose $\gamma^k = \frac{2\alpha}{\alpha^2 k + 2}$ where $\alpha = \frac{\mu_{\mathcal{D}}}{(1-\beta)R_0}$ and run* SMTP *for $k = K$ iterations where*

$$K = \frac{1}{\varepsilon} \cdot \frac{2R_0^2}{\mu_{\mathcal{D}}^2} \max \left\{ (1-\beta)^2 (f(x^0) - f(x^*)), L\gamma_{\mathcal{D}} \right\} - \frac{2(1-\beta)^2 R_0^2}{\mu_{\mathcal{D}}^2}, \qquad \varepsilon > 0, \tag{18}$$

*we get $\mathbf{E}\left[ f(z^K) \right] - f(x^*) \leq \varepsilon$.*

We notice that if we choose $\beta$ sufficiently close to 1, we will obtain from the formula (18) that $K \approx \frac{2R_0^2 L\gamma_{\mathcal{D}}}{\varepsilon\mu_{\mathcal{D}}^2}$.

## 3.3 STRONGLY CONVEX CASE

In this section we present our complexity results for Algorithm 1 when $f$ is $\mu$-strongly convex.

**Assumption 3.3.** *We assume that $f$ is $\mu$-strongly convex with respect to the norm $\| \cdot \|_{\mathcal{D}}^*$:*

$$f(y) \geq f(x) + \langle \nabla f(x), y - x \rangle + \frac{\mu}{2} (\|y - x\|_{\mathcal{D}}^*)^2, \quad \forall x, y \in \mathbb{R}^d. \tag{19}$$

It is well known that strong convexity implies

$$\|\nabla f(x)\|_{\mathcal{D}}^2 \geq 2\mu \left( f(x) - f(x^*) \right). \tag{20}$$

**Theorem 3.4** (Solution-dependent stepsizes). *Let Assumptions 2.1, 3.1 and 3.3 be satisfied. If we set* $\gamma^k = \frac{(1-\beta)\theta_k \mu_{\mathcal{D}}}{L}\sqrt{2\mu(f(z^k) - f(x^*))}$ *for some* $\theta_k \in (0, 2)$ *such that* $\theta = \inf_{k \geq 0}\{2\theta_k - \gamma_{\mathcal{D}}\theta_k^2\} \in (0, L/(\mu_{\mathcal{D}}^2 \mu))$, *then for the iterates of* SMTP, *the following inequality holds:*

$$\mathbf{E}\left[f(z^k)\right] - f(x^*) \leq \left(1 - \frac{\theta\mu_{\mathcal{D}}^2\mu}{L}\right)^k \left(f(x^0) - f(x^*)\right). \tag{21}$$

*Then, If we run* SMTP *for* $k = K$ *iterations where*

$$K = \frac{\kappa}{\theta\mu_{\mathcal{D}}^2}\ln\left(\frac{f(x^0) - f(x^*)}{\varepsilon}\right), \qquad \varepsilon > 0, \tag{22}$$

*where* $\kappa \stackrel{def}{=} \frac{L}{\mu}$ *is the condition number of the objective, we will get* $\mathbf{E}\left[f(z^K)\right] - f(x^*) \leq \varepsilon$.

Note that the previous result uses stepsizes that depends on the optimal solution $f(x^*)$ which is often not known in practice. The next theorem removes this drawback without spoiling the convergence rate. However, we need an additional assumption on the distribution $\mathcal{D}$ and one extra function evaluation.

**Assumption 3.4.** *We assume that for all* $s \sim \mathcal{D}$ *we have* $\|s\|_2 = 1$.

**Theorem 3.5** (Solution-free stepsizes). *Let Assumptions 2.1, 3.1, 3.3 and 3.4 be satisfied. If additionally we compute* $f(z^k + ts^k)$, *set* $\gamma^k = (1-\beta)|f(z^k + ts^k) - f(z^k)|/(Lt)$ *for* $t > 0$ *and assume that* $\mathcal{D}$ *is such that* $\mu_{\mathcal{D}}^2 \leq L/\mu$, *then for the iterates of* SMTP *the following inequality holds:*

$$\mathbf{E}\left[f(z^k)\right] - f(x^*) \leq \left(1 - \frac{\mu_{\mathcal{D}}^2\mu}{L}\right)^k \left(f(x^0) - f(x^*)\right) + \frac{L^2 t^2}{8\mu_{\mathcal{D}}^2\mu}. \tag{23}$$

*Moreover, for any* $\varepsilon > 0$ *if we set* $t$ *such that*

$$0 < t \leq \sqrt{\frac{4\varepsilon\mu_{\mathcal{D}}^2\mu}{L^2}}, \tag{24}$$

*and run* SMTP *for* $k = K$ *iterations where*

$$K = \frac{\kappa}{\mu_{\mathcal{D}}^2}\ln\left(\frac{2(f(x^0) - f(x^*))}{\varepsilon}\right), \tag{25}$$

*where* $\kappa \stackrel{def}{=} \frac{L}{\mu}$ *is the condition number of* $f$, *we will have* $\mathbf{E}\left[f(z^K)\right] - f(x^*) \leq \varepsilon$.

# 4 STOCHASTIC MOMENTUM THREE POINTS WITH IMPORTANCE SAMPLING (SMTP_IS)

In this section we consider another assumption, in a similar spirit to Bibi et al. (2019), on the objective.

**Assumption 4.1** (Coordinate-wise $L$-smoothness). *We assume that the objective* $f$ *has coordinate-wise Lipschitz gradient, with Lipschitz constants* $L_1, \ldots, L_d > 0$, *i.e.*

$$f(x + he_i) \leq f(x) + \nabla_i f(x)h + \frac{L_i}{2}h^2, \qquad \forall x \in \mathbb{R}^d, h \in \mathbb{R}, \tag{26}$$

*where* $\nabla_i f(x)$ *is* $i$-*th partial derivative of* $f$ *at the point* $x$.

For this kind of problems we modify SMTP and present STMP_IS method in Algorithm 2. In general, the idea behind methods with importance sampling and, in particular, behind SMTP_IS is to adjust probabilities of sampling in such a way that gives better convergence guarantees. In the case when $f$ satisfies coordinate-wise $L$-smoothness and Lipschitz constants $L_i$ are known it is natural to sample direction $s^k = e_i$ with probability depending on $L_i$ (e.g. proportional to $L_i$). One can find more detailed discussion of the importance sampling in Zhao & Zhang (2015) and Richtárik & Takáč (2016).

Now, we establish the key result which will be used to prove the main complexity results of STMP_IS.

---

**Algorithm 2** `SMTP_IS`: Stochastic Momentum Three Points with Importance Sampling

---

**Require:** stepsize parameters $w_1, \ldots, w_n > 0$, probabilities $p_1, \ldots, p_n > 0$ summing to 1, starting point $x^0 \in \mathbb{R}^d$, $0 \le \beta < 1$ — momentum parameter
1: Set $v^{-1} = 0$ and $z^0 = x^0$
2: **for** $k = 0, 1, \ldots$ **do**
3:     Select $i_k = i$ with probability $p_i > 0$
4:     Choose stepsize $\gamma_i^k$ proportional to $\frac{1}{w_{i_k}}$
5:     Let $v_+^k = \beta v^{k-1} + e_{i_k}$ and $v_-^k = \beta v^{k-1} - e_{i_k}$
6:     Let $x_+^{k+1} = x^k - \gamma_i^k v_+^k$ and $x_-^{k+1} = x^k - \gamma_i^k v_-^k$
7:     Let $z_+^{k+1} = x_+^{k+1} - \frac{\gamma_i^k \beta}{1-\beta} v_+^k$ and $z_-^{k+1} = x_-^{k+1} - \frac{\gamma_i^k \beta}{1-\beta} v_-^k$
8:     Set $z^{k+1} = \arg\min \left\{ f(z^k), f(z_+^{k+1}), f(z_-^{k+1}) \right\}$
9:     Set $x^{k+1} = \begin{cases} x_+^{k+1}, & \text{if } z^{k+1} = z_+^{k+1} \\ x_-^{k+1}, & \text{if } z^{k+1} = z_-^{k+1} \\ x^k, & \text{if } z^{k+1} = z^k \end{cases}$ and $v^{k+1} = \begin{cases} v_+^{k+1}, & \text{if } z^{k+1} = z_+^{k+1} \\ v_-^{k+1}, & \text{if } z^{k+1} = z_-^{k+1} \\ v^k, & \text{if } z^{k+1} = z^k \end{cases}$
10: **end for**

---

**Lemma 4.1.** *Assume that $f$ satisfies Assumption 4.1. Then for the iterates of* `SMTP_IS` *the following inequalities hold:*

$$f(z^{k+1}) \le f(z^k) - \frac{\gamma_i^k}{1-\beta} |\nabla_{i_k} f(z^k)| + \frac{L_{i_k}(\gamma_i^k)^2}{2(1-\beta)^2} \tag{27}$$

*and*

$$\mathbf{E}_{s^k \sim \mathcal{D}} \left[ f(z^{k+1}) \right] \le f(z^k) - \frac{1}{1-\beta} \mathbf{E} \left[ \gamma_i^k |\nabla_{i_k} f(z^k)| \mid z^k \right] + \frac{1}{2(1-\beta)^2} \mathbf{E} \left[ L_{i_k}(\gamma_i^k)^2 \mid z^k \right]. \tag{28}$$

Due to the page limitation, we provide the complexity results of `SMTP_IS` in the Appendix.

## 5 EXPERIMENTS

**Experimental Setup.** We conduct extensive experiments[3] on challenging non-convex problems on the continuous control task from the MuJoCO suit Todorov et al. (2012). In particular, we address the problem of model-free control of a dynamical system. Policy gradient methods for model-free reinforcement learning algorithms provide an off-the-shelf model-free approach to learn how to control a dynamical system and are often benchmarked in a simulator. We compare our proposed momentum stochastic three points method `SMTP` and the momentum with importance sampling version `SMTP_IS` against state-of-art DFO based methods as `STP_IS` Bibi et al. (2019) and ARS Mania et al. (2018). Moreover, we also compare against classical policy gradient methods as TRPO Schulman et al. (2015) and NG Rajeswaran et al. (2017). We conduct experiments on several environments with varying difficulty `Swimmer-v1`, `Hopper-v1`, `HalfCheetah-v1`, `Ant-v1`, and `Humanoid-v1`.

Note that due to the stochastic nature of problem where $f$ is stochastic, we use the mean of the function values of $f(x^k)$, $f(x_+^k)$ and $f(x_-^k)$, see Algorithm 1, over K observations. Similar to the work in Bibi et al. (2019), we use $K = 2$ for `Swimmer-v1`, $K = 4$ for both `Hopper-v1` and `HalfCheetah-v1`, $K = 40$ for `Ant-v1` and `Humanoid-v1`. Similar to Bibi et al. (2019), these values were chosen based on the validation performance over the grid that is $K \in \{1, 2, 4, 8, 16\}$ for the smaller dimensional problems `Swimmer-v1`, `Hopper-v1`, `HalfCheetah-v1` and $K \in \{20, 40, 80, 120\}$ for larger dimensional problems `Ant-v1`, and `Humanoid-v1`. As for the momentum term, for `SMTP` we set $\beta = 0.5$. For `SMTP_IS`, as the smoothness constants are not available for continuous control, we use the coordinate smoothness constants of a $\theta$ parameterized smooth function $\hat{f}_\theta$ (multi-layer perceptron) that estimates $f$. In particular, consider running any DFO for n steps; with the queried sampled $\{x_i, f(x_i)\}_{i=1}^n$, we estimate $f$ by solving $\theta_{n+1} = \operatorname{argmin}_\theta \sum_i (f(x_i) - \hat{f}(x_i; \theta))^2$. See Bibi et al. (2019) for further implementation details

---

[3]The code will be made available online upon acceptance of this work.

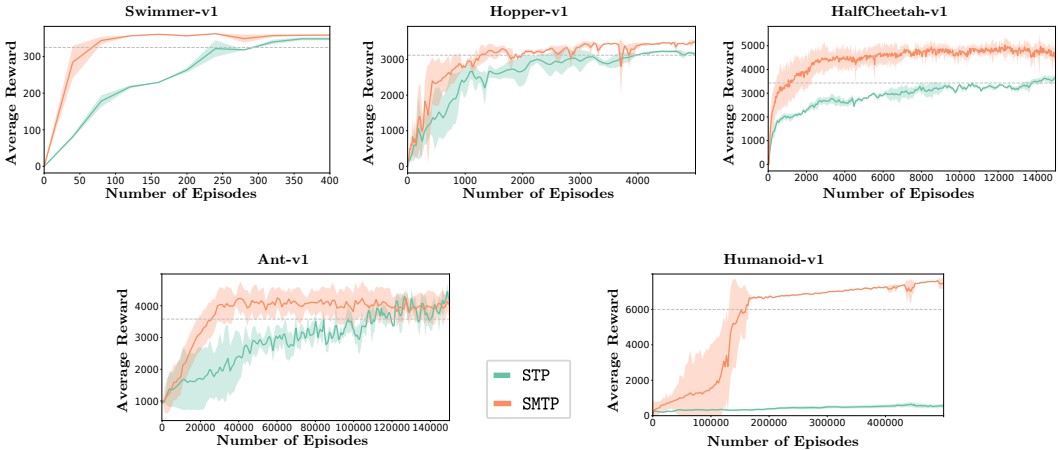

Figure 1: SMTP is far superior to STP on all 5 different MuJoCo tasks particularly on the high dimensional Humanoid-v1 problem. The horizontal dashed lines are the thresholds used in Table 2 to demonstrate complexity of each method.

Table 2: For each MuJoCo task, we report the average number of episodes required to achieve a predefined reward threshold. Results for our method is averaged over five random seeds, the rest is copied from (Mania et al., 2018) (N/A means the method failed to reach the threshold. UNK means the results is unknown since they are not reported in the literature.)

|  | Threshold | STP | STP$_{IS}$ | SMTP | SMTP$_{IS}$ | ARS(V1-t) | ARS(V2-t) | NG-lin | TRPO-nn |
|---|---|---|---|---|---|---|---|---|---|
| Swimmer-v1 | 325 | 320 | 110 | 80 | 100 | 100 | 427 | 1450 | N/A |
| Hopper-v1 | 3120 | 3970 | 2400 | 1264 | 1408 | 51840 | 1973 | 13920 | 10000 |
| HalfCheetah-v1 | 3430 | 13760 | 4420 | 1872 | 1624 | 8106 | 1707 | 11250 | 4250 |
| Ant-v1 | 3580 | 107220 | 43860 | 19890 | 14420 | 58133 | 20800 | 39240 | 73500 |
| Humanoid-v1 | 6000 | N/A | 530200 | 161230 | 207160 | N/A | 142600 | 130000 | UNK |

as we follow the same experimental procedure. In contrast to STP_IS, our method (SMTP) does not required sampling from directions in the canonical basis; hence, we use directions from standard Normal distribution in each iteration. For SMTP_IS, we follow a similar procedure as Bibi et al. (2019) and sample from columns of a random matrix $B$.

Similar to the standard practice, we perform all experiments with 5 different initialization and measure the average reward, in continuous control we are maximizing the reward function $f$, and best and worst run per iteration. We compare algorithms in terms of reward vs. sample complexity.

**Comparison Against** STP. Our method improves sample complexity of STP and STP_IS significantly. Especially for high dimensional problems like Ant-v1 and Humanoid-v1, sample efficiency of SMTP is at least as twice as the STP. Moreover, SMTP_IS helps in some experiments by improving over SMTP. However, this is not consistent in all environments. We believe this is largely due to the fact that SMTP_IS can only handle sampling from canonical basis similar to STP_IS.

**Comparison Against State-of-The-Art**. We compare our method with state-of-the-art DFO and policy gradient algorithms. For the environments, Swimmer-v1, Hopper-v1, HalfCheetah-v1 and Ant-v1, our method outperforms the state-of-the-art results. Whereas for Humanoid-v1, our methods results in a comparable sample complexity.

# 6 CONCLUSION

We have proposed, SMTP, the first heavy ball momentum DFO based algorithm with convergence rates for non-convex, convex and strongly convex functions under generic sampling direction. We specialize the sampling to the set of coordinate bases and further improve rates by proposing a momentum and importance sampling version SMPT_IS with new convergence rates for non-convex, convex and strongly convex functions too. We conduct large number of experiments on the task of

controlling dynamical systems. We outperform two different policy gradient methods and achieve comparable or better performance to the best DFO algorithm (ARS) on the respective environments.

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

# A Stochastic Derivative Free Optimization Method with Momentum

# (Supplementary Material)

## A  PRELIMINARIES

We first list the main assumptions.

**Assumption A.1.** *(L-smoothness) We say that $f$ is L-smooth if:*

$$\|\nabla f(x) - \nabla f(y)\|_2 \le L\|x - y\|_2 \quad \forall x, y \in \mathbb{R}^d. \tag{29}$$

**Assumption A.2.** *The probability distribution $\mathcal{D}$ on $\mathbb{R}^d$ satisfies the following properties:*

1. *The quantity $\gamma_{\mathcal{D}} \stackrel{def}{=} \mathbf{E}_{s\sim\mathcal{D}}\|s\|_2^2$ is positive and finite.*

2. *There is a constant $\mu_{\mathcal{D}} > 0$ and norm $\|\cdot\|_{\mathcal{D}}$ on $\mathbb{R}^d$ such that for all $g \in \mathbb{R}^d$*

$$\mathbf{E}_{s\sim\mathcal{D}}|\langle g, s\rangle| \ge \mu_{\mathcal{D}}\|g\|_{\mathcal{D}}. \tag{30}$$

We establish the key lemma which will be used to prove the theorems stated in the paper.

**Lemma A.1.** *Assume that $f$ is L-smooth and $\mathcal{D}$ satisfies Assumption A.2. Then for the iterates of* `SMTP` *the following inequalities hold:*

$$f(z^{k+1}) \le f(z^k) - \frac{\gamma^k}{1-\beta}|\langle \nabla f(z^k), s^k\rangle| + \frac{L(\gamma^k)^2}{2(1-\beta)^2}\|s^k\|_2^2 \tag{31}$$

*and*

$$\mathbf{E}_{s^k\sim\mathcal{D}}\left[f(z^{k+1})\right] \le f(z^k) - \frac{\gamma^k\mu_{\mathcal{D}}}{1-\beta}\|\nabla f(z^k)\|_{\mathcal{D}} + \frac{L(\gamma^k)^2\gamma_{\mathcal{D}}}{2(1-\beta)^2}. \tag{32}$$

*Proof.* By induction one can show that

$$z^k = x^k - \frac{\gamma^k\beta}{1-\beta}v^{k-1}. \tag{33}$$

That is, for $k = 0$ this recurrence holds and update rules for $z^k, x^k$ and $v^{k-1}$ do not brake it. From this we get

$$
\begin{aligned}
z_+^{k+1} &= x_+^{k+1} - \frac{\gamma^k\beta}{1-\beta}v_+^k = x^k - \gamma^k v_+^k - \frac{\gamma^k\beta}{1-\beta}v_+^k \\
&= x^k - \frac{\gamma^k}{1-\beta}v_+^k = x^k - \frac{\gamma^k\beta}{1-\beta}v^{k-1} - \frac{\gamma^k}{1-\beta}s^k \\
&\stackrel{(33)}{=} z^k - \frac{\gamma^k}{1-\beta}s^k.
\end{aligned}
$$

Similarly,

$$
\begin{aligned}
z_-^{k+1} &= x_-^{k+1} - \frac{\gamma^k\beta}{1-\beta}v_-^k = x^k - \gamma^k v_-^k - \frac{\gamma^k\beta}{1-\beta}v_-^k \\
&= x^k - \frac{\gamma^k}{1-\beta}v_-^k = x^k - \frac{\gamma^k\beta}{1-\beta}v^{k-1} + \frac{\gamma^k}{1-\beta}s^k \\
&\stackrel{(33)}{=} z^k + \frac{\gamma^k}{1-\beta}s^k.
\end{aligned}
$$

It implies that

$$
\begin{aligned}
f(z_+^{k+1}) &\overset{(3)}{\leq} f(z^k) + \langle \nabla f(z^k), z_+^{k+1} - z_k \rangle + \frac{L}{2}\|z_+^{k+1} - z^k\|_2^2 \\
&= f(z^k) - \frac{\gamma^k}{1-\beta}\langle \nabla f(z^k), s^k \rangle + \frac{L(\gamma^k)^2}{2(1-\beta)^2}\|s^k\|_2^2
\end{aligned}
$$

and

$$
f(z_-^{k+1}) \leq f(z^k) + \frac{\gamma^k}{1-\beta}\langle \nabla f(z^k), s^k \rangle + \frac{L(\gamma^k)^2}{2(1-\beta)^2}\|s^k\|_2^2.
$$

Unifying these two inequalities we get

$$
f(z^{k+1}) \leq \min\{f(z_+^{k+1}), f(z_-^{k+1})\} = f(z^k) - \frac{\gamma^k}{1-\beta}|\langle \nabla f(z^k), s^k \rangle| + \frac{L(\gamma^k)^2}{2(1-\beta)^2}\|s^k\|_2^2,
$$

which proves (31). Finally, taking the expectation $\mathbf{E}_{s^k \sim \mathcal{D}}$ of both sides of the previous inequality and invoking Assumption A.2, we obtain

$$
\mathbf{E}_{s^k \sim \mathcal{D}}\left[f(z^{k+1})\right] \leq f(z^k) - \frac{\gamma^k \mu_\mathcal{D}}{1-\beta}\|\nabla f(z^k)\|_\mathcal{D} + \frac{L(\gamma^k)^2 \gamma_\mathcal{D}}{2(1-\beta)^2}.
$$

$\square$

## B  NON-CONVEX CASE

**Theorem B.1.** *Let Assumptions A.1 and A.2 be satisfied. Let* SMTP *with* $\gamma^k \equiv \gamma > 0$ *produce points* $\{z^0, z^1, \ldots, z^{K-1}\}$ *and* $\overline{z}^K$ *is chosen uniformly at random among them. Then*

$$
\mathbf{E}\left[\|\nabla f(\overline{z}^K)\|_\mathcal{D}\right] \leq \frac{(1-\beta)(f(x^0) - f(x^*))}{K\gamma\mu_\mathcal{D}} + \frac{L\gamma\gamma_\mathcal{D}}{2\mu_\mathcal{D}(1-\beta)}. \tag{34}
$$

*Moreover, if we choose* $\gamma = \frac{\gamma_0}{\sqrt{K}}$ *the complexity* (34) *reduces to*

$$
\mathbf{E}\left[\|\nabla f(\overline{z}^K)\|_\mathcal{D}\right] \leq \frac{1}{\sqrt{K}}\left(\frac{(1-\beta)(f(z^0) - f(x^*))}{\gamma_0\mu_\mathcal{D}} + \frac{L\gamma_0\gamma_\mathcal{D}}{2\mu_\mathcal{D}(1-\beta)}\right). \tag{35}
$$

*Then* $\gamma_0 = \sqrt{\frac{2(1-\beta)^2(f(x^0)-f(x^*))}{L\gamma_\mathcal{D}}}$ *minimizes the right-hand side of* (35) *and for this choice we have*

$$
\mathbf{E}\left[\|\nabla f(\overline{z}^K)\|_\mathcal{D}\right] \leq \frac{\sqrt{2\left(f(x^0) - f(x^*)\right)L\gamma_\mathcal{D}}}{\mu_\mathcal{D}\sqrt{K}}. \tag{36}
$$

*Proof.* Taking full expectation from both sides of inequality (32) we get

$$
\mathbf{E}\left[\|\nabla f(z^k)\|_\mathcal{D}\right] \leq \frac{(1-\beta)\mathbf{E}\left[f(z^k) - f(z^{k+1})\right]}{\gamma\mu_\mathcal{D}} + \frac{L\gamma\gamma_\mathcal{D}}{2\mu_\mathcal{D}(1-\beta)}.
$$

Further, summing up the results for $k = 0, 1, \ldots, K-1$, dividing both sides of the obtained inequality by $K$ and using tower property of the mathematical expectation we get

$$
\mathbf{E}\left[\|\nabla f(\overline{z}^K)\|_\mathcal{D}\right] = \frac{1}{K}\sum_{k=0}^{K-1}\mathbf{E}\left[\|\nabla f(z^k)\|_\mathcal{D}\right] \leq \frac{(1-\beta)(f(z^0) - f(x^*))}{K\gamma\mu_\mathcal{D}} + \frac{L\gamma\gamma_\mathcal{D}}{2\mu_\mathcal{D}(1-\beta)}.
$$

The last part where $\gamma = \frac{\gamma_0}{\sqrt{K}}$ is straightforward. $\square$

## C  Convex Case

**Assumption C.1.** *We assume that $f$ is convex, has a minimizer $x^*$ and has bounded level set at $x^0$:*

$$R_0 \overset{def}{=} \max\left\{\|x - x^*\|_{\mathcal{D}}^* \mid f(x) \leq f(x^0)\right\} < +\infty, \tag{37}$$

*where $\|\xi\|_{\mathcal{D}}^* \overset{def}{=} \max\left\{\langle \xi, x \rangle \mid \|x\|_{\mathcal{D}} \leq 1\right\}$ defines the dual norm to $\|\cdot\|_{\mathcal{D}}$.*

**Theorem C.1** (Constant stepsize). *Let Assumptions A.1, A.2 and C.1 be satisfied. If we set $\gamma^k \equiv \gamma < \frac{(1-\beta)R_0}{\mu_{\mathcal{D}}}$, then for the iterates of* SMTP *method the following inequality holds:*

$$\mathbf{E}\left[f(z^k) - f(x^*)\right] \leq \left(1 - \frac{\gamma\mu_{\mathcal{D}}}{(1-\beta)R_0}\right)^k \left(f(x^0) - f(x^*)\right) + \frac{L\gamma\gamma_{\mathcal{D}}R_0}{2(1-\beta)\mu_{\mathcal{D}}}. \tag{38}$$

*If we choose $\gamma = \frac{\varepsilon(1-\beta)\mu_{\mathcal{D}}}{L\gamma_{\mathcal{D}}R_0}$ for some $0 < \varepsilon \leq \frac{L\gamma_{\mathcal{D}}R_0^2}{\mu_{\mathcal{D}}^2}$ and run* SMTP *for $k = K$ iterations where*

$$K = \frac{1}{\varepsilon}\frac{L\gamma_{\mathcal{D}}R_0^2}{\mu_{\mathcal{D}}^2}\ln\left(\frac{2(f(x^0) - f(x^*))}{\varepsilon}\right), \tag{39}$$

*then we will get $\mathbf{E}\left[f(z^K)\right] - f(x^*) \leq \varepsilon$.*

*Proof.* From the (32) and monotonicity of $\{f(z^k)\}_{k \geq 0}$ we have

$$
\begin{aligned}
\mathbf{E}_{s \sim \mathcal{D}}\left[f(z^{k+1})\right] &\leq f(z^k) - \frac{\gamma\mu_{\mathcal{D}}}{1-\beta}\|\nabla f(z^k)\|_{\mathcal{D}} + \frac{L\gamma^2\gamma_{\mathcal{D}}}{2(1-\beta)^2} \\
&\overset{(14)}{\leq} f(z^k) - \frac{\gamma\mu_{\mathcal{D}}}{(1-\beta)R_0}(f(z^k) - f(x^*)) + \frac{L\gamma^2\gamma_{\mathcal{D}}}{2(1-\beta)^2}.
\end{aligned}
$$

Taking full expectation, subtracting $f(x^*)$ from the both sides of the previous inequality and using the tower property of mathematical expectation we get

$$\mathbf{E}\left[f(z^{k+1}) - f(x^*)\right] \leq \left(1 - \frac{\gamma\mu_{\mathcal{D}}}{(1-\beta)R_0}\right)\mathbf{E}\left[f(z^k) - f(x^*)\right] + \frac{L\gamma^2\gamma_{\mathcal{D}}}{2(1-\beta)^2}. \tag{40}$$

Since $\gamma < \frac{(1-\beta)R_0}{\mu_{\mathcal{D}}}$ the term $1 - \frac{\gamma\mu_{\mathcal{D}}}{(1-\beta)R_0}$ is positive and we can unroll the recurrence (40):

$$
\begin{aligned}
\mathbf{E}\left[f(z^k) - f(x^*)\right] &\leq \left(1 - \frac{\gamma\mu_{\mathcal{D}}}{(1-\beta)R_0}\right)^k \left(f(z^0) - f(x^*)\right) + \frac{L\gamma^2\gamma_{\mathcal{D}}}{2(1-\beta)^2}\sum_{l=0}^{k-1}\left(1 - \frac{\gamma\mu_{\mathcal{D}}}{(1-\beta)R_0}\right)^l \\
&\leq \left(1 - \frac{\gamma\mu_{\mathcal{D}}}{(1-\beta)R_0}\right)^k \left(f(x^0) - f(x^*)\right) + \frac{L\gamma^2\gamma_{\mathcal{D}}}{2(1-\beta)^2}\sum_{l=0}^{\infty}\left(1 - \frac{\gamma\mu_{\mathcal{D}}}{(1-\beta)R_0}\right)^l \\
&\leq \left(1 - \frac{\gamma\mu_{\mathcal{D}}}{(1-\beta)R_0}\right)^k \left(f(x^0) - f(x^*)\right) + \frac{L\gamma^2\gamma_{\mathcal{D}}}{2(1-\beta)^2}\cdot\frac{(1-\beta)R_0}{\gamma\mu_{\mathcal{D}}} \\
&= \left(1 - \frac{\gamma\mu_{\mathcal{D}}}{(1-\beta)R_0}\right)^k \left(f(x^0) - f(x^*)\right) + \frac{L\gamma\gamma_{\mathcal{D}}R_0}{2(1-\beta)\mu_{\mathcal{D}}}.
\end{aligned}
$$

Lastly, putting $\gamma = \frac{\varepsilon(1-\beta)\mu_{\mathcal{D}}}{L\gamma_{\mathcal{D}}R_0}$ and $k = K$ from (39) in (38) we have

$$
\begin{aligned}
\mathbf{E}[f(z^K)] - f(x^*) &= \left(1 - \frac{\varepsilon\mu_{\mathcal{D}}^2}{L\gamma_{\mathcal{D}}R_0^2}\right)^K \left(f(x^0) - f(x^*)\right) + \frac{\varepsilon}{2} \\
&\leq \exp\left\{-K\cdot\frac{\varepsilon\mu_{\mathcal{D}}^2}{L\gamma_{\mathcal{D}}R_0^2}\right\}\left(f(x^0) - f(x^*)\right) + \frac{\varepsilon}{2} \\
&\overset{(39)}{=} \frac{\varepsilon}{2} + \frac{\varepsilon}{2} = \varepsilon.
\end{aligned}
$$

$\square$

Next we use technical lemma from Mishchenko et al. (2019). We provide the original proof for completeness.

**Lemma C.1** (Lemma 6 from Mishchenko et al. (2019)). *Let a sequence $\{a^k\}_{k \geq 0}$ satisfy inequality $a^{k+1} \leq (1 - \gamma^k \alpha)a^k + (\gamma^k)^2 N$ for any positive $\gamma^k \leq \gamma_0$ with some constants $\alpha > 0, N > 0, \gamma_0 > 0$. Further, let $\theta \geq \frac{2}{\gamma_0}$ and take $C$ such that $N \leq \frac{\alpha\theta}{4}C$ and $a_0 \leq C$. Then, it holds*

$$a^k \leq \frac{C}{\frac{\alpha}{\theta}k + 1}$$

*if we set $\gamma^k = \frac{2}{\alpha k + \theta}$.*

*Proof.* We will show the inequality for $a^k$ by induction. Since inequality $a_0 \leq C$ is one of our assumptions, we have the initial step of the induction. To prove the inductive step, consider

$$a^{k+1} \leq (1 - \gamma^k \alpha)a^k + (\gamma^k)^2 N \leq \left(1 - \frac{2\alpha}{\alpha k + \theta}\right)\frac{\theta C}{\alpha k + \theta} + \theta\alpha\frac{C}{(\alpha k + \theta)^2}.$$

To show that the right-hand side is upper bounded by $\frac{\theta C}{\alpha(k+1)+\theta}$, one needs to have, after multiplying both sides by $(\alpha k + \theta)(\alpha k + \alpha + \theta)(\theta C)^{-1}$,

$$\left(1 - \frac{2\alpha}{\alpha k + \theta}\right)(\alpha k + \alpha + \theta) + \alpha\frac{\alpha k + \alpha + \theta}{\alpha k + \theta} \leq \alpha k + \theta,$$

which is equivalent to

$$\alpha - \alpha\frac{\alpha k + \alpha + \theta}{\alpha k + \theta} \leq 0.$$

The last inequality is trivially satisfied for all $k \geq 0$. $\qquad\square$

**Theorem C.2** (Decreasing stepsizes). *Let Assumptions A.1, A.2 and C.1 be satisfied. If we set $\gamma^k = \frac{2}{\alpha k + \theta}$, where $\alpha = \frac{\mu_\mathcal{D}}{(1-\beta)R_0}$ and $\theta \geq \frac{2}{\alpha}$, then for the iterates of $\mathtt{SMTP}$ method the following inequality holds:*

$$\mathbf{E}\left[f(z^k)\right] - f(x^*) \leq \frac{1}{\eta k + 1}\max\left\{f(x^0) - f(x^*), \frac{2L\gamma_\mathcal{D}}{\alpha\theta(1-\beta)^2}\right\}, \qquad (41)$$

*where $\eta \stackrel{def}{=} \frac{\alpha}{\theta}$. Then, if we choose $\gamma^k = \frac{2\alpha}{\alpha^2 k + 2}$ where $\alpha = \frac{\mu_\mathcal{D}}{(1-\beta)R_0}$ and run $\mathtt{SMTP}$ for $k = K$ iterations where*

$$K = \frac{1}{\varepsilon} \cdot \frac{2R_0^2}{\mu_\mathcal{D}^2}\max\left\{(1-\beta)^2(f(x^0) - f(x^*)), L\gamma_\mathcal{D}\right\} - \frac{2(1-\beta)^2 R_0^2}{\mu_\mathcal{D}^2}, \qquad \varepsilon > 0, \qquad (42)$$

*we get $\mathbf{E}\left[f(z^K)\right] - f(x^*) \leq \varepsilon$.*

*Proof.* In (40) we proved that

$$\mathbf{E}\left[f(z^{k+1}) - f(x^*)\right] \leq \left(1 - \frac{\gamma\mu_\mathcal{D}}{(1-\beta)R_0}\right)\mathbf{E}\left[f(z^k) - f(x^*)\right] + \frac{L\gamma^2\gamma_\mathcal{D}}{2(1-\beta)^2}.$$

Having that, we can apply Lemma C.1 to the sequence $\mathbf{E}\left[f(z^k) - f(x^*)\right]$. The constants for the lemma are: $N = \frac{L\gamma_\mathcal{D}}{2(1-\beta)^2}$, $\alpha = \frac{\mu_\mathcal{D}}{(1-\beta)R_0}$ and $C = \max\left\{f(x^0) - f(x^*), \frac{2L\gamma_\mathcal{D}}{\alpha\theta(1-\beta)^2}\right\}$. Lastly, choosing $\gamma^k = \frac{2\alpha}{\alpha^2 k + 2}$ is equivalent to the choice $\theta = \frac{2}{\alpha}$. In this case, we have $\alpha\theta = 2$, $C = \max\left\{f(x^0) - f(x^*), \frac{L\gamma_\mathcal{D}}{(1-\beta)^2}\right\}$ and $\eta = \frac{\alpha}{\theta} = \frac{\alpha^2}{2} = \frac{\mu_\mathcal{D}^2}{2(1-\beta)^2 R_0^2}$. Putting these parameters and $K$ from (42) in the (41) we get the result. $\qquad\square$

## D  STRONGLY CONVEX CASE

**Assumption D.1.** *We assume that $f$ is $\mu$-strongly convex with respect to the norm $\|\cdot\|_{\mathcal{D}}^*$:*

$$f(y) \geq f(x) + \langle \nabla f(x), y - x \rangle + \frac{\mu}{2}(\|y - x\|_{\mathcal{D}}^*)^2, \quad \forall x, y \in \mathbb{R}^d. \tag{43}$$

It is well known that strong convexity implies

$$\|\nabla f(x)\|_{\mathcal{D}}^2 \geq 2\mu \left( f(x) - f(x^*) \right). \tag{44}$$

**Theorem D.1** (Solution-dependent stepsizes). *Let Assumptions A.1, A.2 and D.1 be satisfied. If we set $\gamma^k = \frac{(1-\beta)\theta_k \mu_{\mathcal{D}}}{L}\sqrt{2\mu(f(z^k) - f(x^*))}$ for some $\theta_k \in (0, 2)$ such that $\theta = \inf\limits_{k \geq 0}\{2\theta_k - \gamma_{\mathcal{D}}\theta_k^2\} \in \left(0, \frac{L}{\mu_{\mathcal{D}}^2 \mu}\right)$, then for the iterates of* SMTP *the following inequality holds:*

$$\mathbf{E}\left[f(z^k)\right] - f(x^*) \leq \left(1 - \frac{\theta \mu_{\mathcal{D}}^2 \mu}{L}\right)^k \left(f(x^0) - f(x^*)\right). \tag{45}$$

*If we run* SMTP *for $k = K$ iterations where*

$$K = \frac{\kappa}{\theta \mu_{\mathcal{D}}^2} \ln\left(\frac{f(x^0) - f(x^*)}{\varepsilon}\right), \qquad \varepsilon > 0, \tag{46}$$

*where $\kappa \stackrel{def}{=} \frac{L}{\mu}$ is the condition number of the objective, we will get $\mathbf{E}\left[f(z^K)\right] - f(x^*) \leq \varepsilon$.*

*Proof.* From (32) and $\gamma^k = \frac{\theta_k \mu_{\mathcal{D}}}{L}\sqrt{2\mu(f(x^k) - f(x^*))}$ we have

$$
\begin{aligned}
\mathbf{E}_{s^k \sim \mathcal{D}}\left[f(z^{k+1})\right] - f(x^*) &\leq f(z^k) - f(x^*) - \frac{\gamma^k \mu_{\mathcal{D}}}{1 - \beta}\|\nabla f(z^k)\|_{\mathcal{D}} + \frac{L(\gamma^k)^2 \gamma_{\mathcal{D}}}{2(1-\beta)^2} \\
&\overset{(44)}{\leq} f(z^k) - f(x^*) - \frac{\gamma^k \mu_{\mathcal{D}}}{1 - \beta}\sqrt{2\mu(f(z^k) - f(x^*))} \\
&\quad + \frac{\gamma_{\mathcal{D}}\theta_k^2 \mu_{\mathcal{D}}^2 \mu}{L}(f(z^k) - f(x^*)) \\
&\leq f(z^k) - f(x^*) - \frac{2\theta^k \mu_{\mathcal{D}}^2 \mu}{L}(f(z^k) - f(x^*)) \\
&\quad + \frac{\gamma_{\mathcal{D}}\theta_k^2 \mu_{\mathcal{D}}^2 \mu}{L}(f(z^k) - f(x^*)) \\
&\leq \left(1 - (2\theta_k - \gamma_{\mathcal{D}}\theta_k^2)\frac{\mu_{\mathcal{D}}^2 \mu}{L}\right)(f(z^k) - f(x^*)).
\end{aligned}
$$

Using $\theta = \inf\limits_{k \geq 0}\{2\theta_k - \gamma_{\mathcal{D}}\theta_k^2\} \in \left(0, \frac{L}{\mu_{\mathcal{D}}^2 \mu}\right)$ and taking the full expectation from the previous inequality we get

$$
\begin{aligned}
\mathbf{E}\left[f(z^{k+1}) - f(x^*)\right] &\leq \left(1 - \frac{\theta \mu_{\mathcal{D}}^2 \mu}{L}\right)\mathbf{E}\left[f(z^k) - f(x^*)\right] \\
&\leq \left(1 - \frac{\theta \mu_{\mathcal{D}}^2 \mu}{L}\right)^{k+1}\left(f(x^0) - f(x^*)\right).
\end{aligned}
$$

Lastly, from (45) we have

$$
\begin{aligned}
\mathbf{E}\left[f(z^K)\right] - f(x^*) &\leq \left(1 - \frac{\theta \mu_{\mathcal{D}}^2 \mu}{L}\right)^K \left(f(x^0) - f(x^*)\right) \\
&\leq \exp\left\{-K\frac{\theta \mu_{\mathcal{D}}^2 \mu}{L}\right\}\left(f(x^0) - f(x^*)\right) \\
&\overset{(46)}{\leq} \varepsilon.
\end{aligned}
$$

$\square$

**Assumption D.2.** *We assume that for all $s \sim \mathcal{D}$ we have $\|s\|_2 = 1$.*

**Theorem D.2** (Solution-free stepsizes). *Let Assumptions A.1, A.2, D.1 and D.2 be satisfied. If additionally we compute $f(z^k + ts^k)$, set $\gamma^k = \frac{(1-\beta)|f(z^k+ts^k)-f(z^k)|}{Lt}$ for $t > 0$ and assume that $\mathcal{D}$ is such that $\mu_{\mathcal{D}}^2 \leq \frac{L}{\mu}$, then for the iterates of* SMTP *the following inequality holds:*

$$\mathbf{E}\left[f(z^k)\right] - f(x^*) \leq \left(1 - \frac{\mu_{\mathcal{D}}^2 \mu}{L}\right)^k \left(f(x^0) - f(x^*)\right) + \frac{L^2 t^2}{8\mu_{\mathcal{D}}^2 \mu}. \tag{47}$$

*Moreover, for any $\varepsilon > 0$ if we set $t$ such that*

$$0 < t \leq \sqrt{\frac{4\varepsilon\mu_{\mathcal{D}}^2\mu}{L^2}}, \tag{48}$$

*and run* SMTP *for $k = K$ iterations where*

$$K = \frac{\kappa}{\mu_{\mathcal{D}}^2} \ln\left(\frac{2(f(x^0) - f(x^*))}{\varepsilon}\right), \tag{49}$$

*where $\kappa \overset{def}{=} \frac{L}{\mu}$ is the condition number of $f$, we will have $\mathbf{E}\left[f(z^K)\right] - f(x^*) \leq \varepsilon$.*

*Proof.* Recall that from (31) we have

$$f(z^{k+1}) \leq f(z^k) - \frac{\gamma^k}{1-\beta}|\langle \nabla f(z^k), s^k \rangle| + \frac{L(\gamma^k)^2}{2(1-\beta)^2}.$$

If we minimize the right hand side of the previous inequality as a function of $\gamma^k$, we will get that the optimal choice in this sense is $\gamma_{\text{opt}}^k = \frac{(1-\beta)|\langle \nabla f(z^k), s^k \rangle|}{L}$. However, this stepsize is impractical for derivative-free optimization, since it requires to know $\nabla f(z^k)$. The natural way to handle this is to approximate directional derivative $\langle \nabla f(z^k), s^k \rangle$ by finite difference $\frac{f(z^k+ts^k)-f(z^k)}{t}$ and that is what we do. We choose $\gamma^k = \frac{(1-\beta)|f(z^k+ts^k)-f(z^k)|}{Lt} = \frac{(1-\beta)|\langle \nabla f(z^k), s^k \rangle|}{L} + \frac{(1-\beta)|f(z^k+ts^k)-f(z^k)|}{Lt} - \frac{(1-\beta)|\langle \nabla f(z^k), s^k \rangle|}{L} \overset{def}{=} \gamma_{\text{opt}}^k + \delta^k$. From this we get

$$f(z^{k+1}) \quad \leq \quad f(z^k) - \frac{|\langle \nabla f(z^k), s^k \rangle|^2}{2L} + \frac{L}{2(1-\beta)^2}(\delta^k)^2.$$

Next we estimate $|\delta^k|$:

$$
\begin{aligned}
|\delta^k| \quad &= \quad \frac{(1-\beta)}{Lt}\left| |f(z^k + ts^k) - f(z^k)| - |\langle \nabla f(z^k), ts^k \rangle| \right| \\
&\leq \quad \frac{(1-\beta)}{Lt}\left| f(z^k + ts^k) - f(z^k) - \langle \nabla f(z^k), ts^k \rangle \right| \\
&\overset{(3)}{\leq} \quad \frac{(1-\beta)}{Lt} \cdot \frac{L}{2}\|ts^k\|_2^2 = \frac{(1-\beta)t}{2}.
\end{aligned}
$$

It implies that

$$
\begin{aligned}
f(z^{k+1}) \quad &\leq \quad f(z^k) - \frac{|\langle \nabla f(z^k), s^k \rangle|^2}{2L} + \frac{L}{2(1-\beta)^2} \cdot \frac{(1-\beta)^2 t^2}{4} \\
&= \quad f(z^k) - \frac{|\langle \nabla f(z^k), s^k \rangle|^2}{2L} + \frac{Lt^2}{8}
\end{aligned}
$$

and after taking full expectation from the both sides of the obtained inequality we get

$$\mathbf{E}\left[f(z^{k+1}) - f(x^*)\right] \leq \mathbf{E}\left[f(z^k) - f(x^*)\right] - \frac{1}{2L}\mathbf{E}\left[|\langle \nabla f(z^k), s^k \rangle|^2\right] + \frac{Lt^2}{8}.$$

Note that from the tower property of mathematical expectation and Jensen's inequality we have

$$
\begin{aligned}
\mathbf{E}\left[|\langle \nabla f(z^k), s^k \rangle|^2\right] \quad &= \quad \mathbf{E}\left[\mathbf{E}_{s^k \sim \mathcal{D}}\left[|\langle \nabla f(z^k), s^k \rangle|^2 \mid z^k\right]\right] \\
&\geq \quad \mathbf{E}\left[\left(\mathbf{E}_{s^k \sim \mathcal{D}}\left[|\langle \nabla f(z^k), s^k \rangle| \mid z^k\right]\right)^2\right] \\
&\overset{(30)}{\geq} \quad \mathbf{E}\left[\mu_{\mathcal{D}}^2\|\nabla f(z^k)\|_{\mathcal{D}}^2\right] \overset{(44)}{\geq} 2\mu_{\mathcal{D}}^2\mu\mathbf{E}\left[f(z^k) - f(x^*)\right].
\end{aligned}
$$

Putting all together we get

$$\mathbf{E}\left[f(z^{k+1}) - f(x^*)\right] \le \left(1 - \frac{\mu_{\mathcal{D}}^2 \mu}{L}\right) \mathbf{E}\left[f(z^k) - f(x^*)\right] + \frac{Lt^2}{8}.$$

Due to $\mu_{\mathcal{D}}^2 \le \frac{L}{\mu}$ we have

$$
\begin{aligned}
\mathbf{E}\left[f(z^k) - f(x^*)\right] &\le \left(1 - \frac{\mu_{\mathcal{D}}^2 \mu}{L}\right)^k \left(f(x^0) - f(x^*)\right) + \frac{Lt^2}{8} \sum_{l=0}^{k-1} \left(1 - \frac{\mu_{\mathcal{D}}^2 \mu}{L}\right)^l \\
&\le \left(1 - \frac{\mu_{\mathcal{D}}^2 \mu}{L}\right)^k \left(f(x^0) - f(x^*)\right) + \frac{Lt^2}{8} \sum_{l=0}^{\infty} \left(1 - \frac{\mu_{\mathcal{D}}^2 \mu}{L}\right)^l \\
&= \left(1 - \frac{\mu_{\mathcal{D}}^2 \mu}{L}\right)^k \left(f(x^0) - f(x^*)\right) + \frac{L^2 t^2}{8\mu_{\mathcal{D}}^2 \mu}.
\end{aligned}
$$

Lastly, from (47) we have

$$
\begin{aligned}
\mathbf{E}\left[f(z^K)\right] - f(x^*) &\le \left(1 - \frac{\mu_{\mathcal{D}}^2 \mu}{L}\right)^K \left(f(x^0) - f(x^*)\right) + \frac{L^2 t^2}{8\mu_{\mathcal{D}}^2 \mu} \\
&\overset{(48)}{\le} \exp\left\{-K\frac{\mu_{\mathcal{D}}^2 \mu}{L}\right\} \left(f(x^0) - f(x^*)\right) + \frac{\varepsilon}{2} \\
&\overset{(49)}{\le} \frac{\varepsilon}{2} + \frac{\varepsilon}{2} = \varepsilon.
\end{aligned}
$$

$\square$

# E    SMTP_IS: STOCHASTIC MOMENTUM THREE POINTS WITH IMPORTANCE SAMPLING

Again by definition of $z^{k+1}$ we get that the sequence $\{f(z^k)\}_{k\geq 0}$ is monotone:

$$f(z^{k+1}) \leq f(z^k) \qquad \forall k \geq 0. \tag{50}$$

**Lemma E.1.** *Assume that $f$ satisfies Assumption 4.1. Then for the iterates of* SMTP_IS *the following inequalities hold:*

$$f(z^{k+1}) \leq f(z^k) - \frac{\gamma_i^k}{1-\beta}|\nabla_{i_k}f(z^k)| + \frac{L_{i_k}(\gamma_i^k)^2}{2(1-\beta)^2} \tag{51}$$

*and*

$$\mathbf{E}_{s^k \sim \mathcal{D}}\left[f(z^{k+1})\right] \leq f(z^k) - \frac{1}{1-\beta}\mathbf{E}\left[\gamma_i^k|\nabla_{i_k}f(z^k)|\mid z^k\right] + \frac{1}{2(1-\beta)^2}\mathbf{E}\left[L_{i_k}(\gamma_i^k)^2 \mid z^k\right]. \tag{52}$$

*Proof.* In the similar way as in Lemma A.1 one can show that

$$z^k = x^k - \frac{\gamma_i^k\beta}{1-\beta}v^{k-1} \tag{53}$$

and

$$z_+^{k+1} = z^k - \frac{\gamma_i^k}{1-\beta}e_{i_k},$$

$$z_-^{k+1} = z^k + \frac{\gamma_i^k}{1-\beta}e_{i_k}.$$

It implies that

$$f(z_+^{k+1}) \overset{(26)}{\leq} f(z^k) - \frac{\gamma_i^k}{1-\beta}\nabla_i f(z^k) + \frac{L_{i_k}(\gamma_i^k)^2}{2(1-\beta)^2}$$

and

$$f(z_-^{k+1}) \leq f(z^k) + \frac{\gamma_i^k}{1-\beta}\nabla_i f(z^k) + \frac{L_{i_k}(\gamma_i^k)^2}{2(1-\beta)^2}.$$

Unifying these two inequalities we get

$$f(z^{k+1}) \leq \min\{f(z_+^{k+1}), f(z_-^{k+1})\} = f(z^k) - \frac{\gamma_i^k}{1-\beta}|\nabla_i f(z^k)| + \frac{L_{i_k}(\gamma_i^k)^2}{2(1-\beta)^2},$$

which proves (51). Finally, taking the expectation $\mathbf{E}[\cdot \mid z^k]$ conditioned on $z^k$ from the both sides of the previous inequality we obtain

$$\mathbf{E}\left[f(z^{k+1}) \mid z^k\right] \leq f(z^k) - \frac{1}{1-\beta}\mathbf{E}\left[\gamma_i^k|\nabla_{i_k}f(z^k)| \mid z^k\right] + \frac{1}{2(1-\beta)^2}\mathbf{E}\left[L_{i_k}(\gamma_i^k)^2 \mid z^k\right].$$

□

## E.1    NON-CONVEX CASE

**Theorem E.1.** *Assume that $f$ satisfies Assumption 4.1. Let* SMTP_IS *with $\gamma_i^k = \frac{\gamma}{w_{i_k}}$ for some $\gamma > 0$ produce points $\{z^0, z^1, \ldots, z^{K-1}\}$ and $\overline{z}^K$ is chosen uniformly at random among them. Then*

$$\mathbf{E}\left[\|\nabla f(\overline{z}^K)\|_1\right] \leq \frac{(1-\beta)(f(x^0) - f(x^*))}{K\gamma \min\limits_{i=1,\ldots,d}\frac{p_i}{w_i}} + \frac{\gamma}{2(1-\beta)\min\limits_{i=1,\ldots,d}\frac{p_i}{w_i}}\sum_{i=1}^{d}\frac{L_i p_i}{w_i^2}. \tag{54}$$

*Moreover, if we choose $\gamma = \frac{\gamma_0}{\sqrt{K}}$, then*

$$\mathbf{E}\left[\|\nabla f(\overline{z}^K)\|_1\right] \leq \frac{1}{\sqrt{K} \min\limits_{i=1,\dots,d} \frac{p_i}{w_i}} \left(\frac{(1-\beta)(f(x^0) - f(x^*))}{\gamma_0} + \frac{\gamma_0}{2(1-\beta)} \sum_{i=1}^{d} \frac{L_i p_i}{w_i^2}\right). \quad (55)$$

*Note that if we choose $\gamma_0 = \sqrt{\frac{2(1-\beta)^2(f(x^0)-f(x^*))}{\sum\limits_{i=1}^{d} \frac{L_i p_i}{w_i^2}}}$ in order to minimize right-hand side of* (55), *we will get*

$$\mathbf{E}\left[\|\nabla f(\overline{z}^K)\|_1\right] \leq \frac{\sqrt{2\left(f(x^0) - f(x^*)\right) \sum\limits_{i=1}^{d} \frac{L_i p_i}{w_i^2}}}{\sqrt{K} \min\limits_{i=1,\dots,d} \frac{p_i}{w_i}}. \quad (56)$$

*Note that for $p_i = L_i / \sum_i^d L_i$ with $w_i = L_i$ we have that the rates improves to*

$$\mathbf{E}\left[\|\nabla f(\overline{z}^K)\|_1\right] \leq \frac{\sqrt{2(f(x^0) - f(x^*))d \sum_{i=1}^{d} L_i}}{\sqrt{K}}. \quad (57)$$

*Proof.* Recall that from (52) we have

$$\mathbf{E}\left[f(z^{k+1}) \mid z^k\right] \leq f(z^k) - \frac{1}{1-\beta}\mathbf{E}\left[\gamma_i^k |\nabla_{i_k} f(z^k)| \mid z^k\right] + \frac{1}{2(1-\beta)^2}\mathbf{E}\left[L_{i_k}(\gamma_i^k)^2 \mid z^k\right]. \quad (58)$$

Using our choice $\gamma_i^k = \frac{\gamma}{w_{i_k}}$ we derive

$$\mathbf{E}\left[\gamma_i^k |\nabla_{i_k} f(z^k)| \mid z^k\right] = \gamma \sum_{i=1}^{d} \frac{p_i}{w_i}|\nabla_i f(z^k)| \geq \gamma\|\nabla f(z^k)\|_1 \min_{i=1,\dots,d} \frac{p_i}{w_i}$$

and

$$\mathbf{E}\left[L_{i_k}(\gamma_i^k)^2 \mid z^k\right] = \gamma^2 \sum_{i=1}^{d} \frac{L_i p_i}{w_i^2}.$$

Putting it in (58) and taking full expectation from the both sides of obtained inequality we get

$$\mathbf{E}\left[f(z^{k+1})\right] \leq \mathbf{E}\left[f(z^k)\right] - \frac{\gamma \min\limits_{i=1,\dots,d} \frac{p_i}{w_i}}{1-\beta}\mathbf{E}\|\nabla f(z^k)\|_1 + \frac{\gamma^2}{2(1-\beta)^2} \sum_{i=1}^{d} \frac{L_i p_i}{w_i^2},$$

whence

$$\|\nabla f(z^k)\|_1 \leq \frac{(1-\beta)\left(\mathbf{E}\left[f(z^k)\right] - \mathbf{E}\left[f(z^{k+1})\right]\right)}{\gamma \min\limits_{i=1,\dots,d} \frac{p_i}{w_i}} + \frac{\gamma}{2(1-\beta) \min\limits_{i=1,\dots,d} \frac{p_i}{w_i}} \sum_{i=1}^{d} \frac{L_i p_i}{w_i^2}.$$

Summing up previous inequality for $k = 0, 1, \dots, K-1$ and dividing both sides of the result by $K$, we get

$$\frac{1}{K}\sum_{k=0}^{K-1} \mathbf{E}\left[\|\nabla f(z^k)\|_1\right] \leq \frac{(1-\beta)(f(z^0) - f(x^*))}{K\gamma \min\limits_{i=1,\dots,d} \frac{p_i}{w_i}} + \frac{\gamma}{2(1-\beta) \min\limits_{i=1,\dots,d} \frac{p_i}{w_i}} \sum_{i=1}^{d} \frac{L_i p_i}{w_i^2}.$$

It remains to notice that $\frac{1}{K}\sum\limits_{k=0}^{K-1} \mathbf{E}\left[\|\nabla f(z^k)\|_1\right] = \mathbf{E}\left[\|\nabla f(\overline{z}^K)\|_1\right]$. The last part where $\gamma = \frac{\gamma_0}{\sqrt{K}}$ is straightforward. $\qquad \square$

### E.2 CONVEX CASE

As for SMTP to tackle convex problems by SMTP_IS we use Assumption 3.2 with $\|\cdot\|_{\mathcal{D}} = \|\cdot\|_1$. Note that in this case $R_0 = \max\left\{\|x - x^*\|_\infty \mid f(x) \le f(x^0)\right\}$.

**Theorem E.2** (Constant stepsize). *Let Assumptions 3.2 and 4.1 be satisfied. If we set $\gamma_i^k = \frac{\gamma}{w_{i_k}}$ such that $0 < \gamma \le \frac{(1-\beta)R_0}{\min\limits_{i=1,\ldots,d} \frac{p_i}{w_i}}$, then for the iterates of SMTP_IS method the following inequality holds:*

$$\mathbf{E}\left[f(z^k) - f(x^*)\right] \le \left(1 - \frac{\gamma \min\limits_{i=1,\ldots,d} \frac{p_i}{w_i}}{(1-\beta)R_0}\right)^k \left(f(z^0) - f(x^*)\right) + \frac{\gamma R_0}{2(1-\beta)\min\limits_{i=1,\ldots,d} \frac{p_i}{w_i}} \sum_{i=1}^d \frac{L_i p_i}{w_i^2}. \tag{59}$$

*Moreover, if we choose $\gamma = \frac{\varepsilon(1-\beta)\min\limits_{i=1,\ldots,d}\frac{p_i}{w_i}}{R_0 \sum\limits_{i=1}^d \frac{L_i p_i}{w_i^2}}$ for some $0 < \varepsilon \le \frac{R_0^2 \sum\limits_{i=1}^d \frac{L_i p_i}{w_i^2}}{\min\limits_{i=1,\ldots,d}\frac{p_i^2}{w_i^2}}$ and run SMTP_IS for $k = K$ iterations where*

$$K = \frac{1}{\varepsilon} \frac{R_0^2 \sum\limits_{i=1}^d \frac{L_i p_i}{w_i^2}}{\min\limits_{i=1,\ldots,d}\frac{p_i^2}{w_i^2}} \ln\left(\frac{2(f(x^0) - f(x^*))}{\varepsilon}\right), \tag{60}$$

*we will get $\mathbf{E}\left[f(z^K)\right] - f(x^*) \le \varepsilon$. Moreover, for $p_i = L_i/\sum_i^d L_i$ with $w_i = L_i$, the rate improves to*

$$K = \frac{1}{\varepsilon} R_0^2 d \sum_{i=1}^d L_i \ln\left(\frac{2(f(x^0) - f(x^*))}{\varepsilon}\right). \tag{61}$$

*Proof.* Recall that from (52) we have

$$\mathbf{E}\left[f(z^{k+1}) \mid z^k\right] \le f(z^k) - \frac{1}{1-\beta}\mathbf{E}\left[\gamma_i^k |\nabla_{i_k} f(z^k)| \mid z^k\right] + \frac{1}{2(1-\beta)^2}\mathbf{E}\left[L_{i_k}(\gamma_i^k)^2 \mid z^k\right]. \tag{62}$$

Using our choice $\gamma_i^k = \frac{\gamma}{w_{i_k}}$ we derive

$$
\begin{aligned}
\mathbf{E}\left[\gamma_i^k \nabla_{i_k} f(z^k) \mid z^k\right] &= \gamma \sum_{i=1}^d \frac{p_i}{w_i}|\nabla_i f(z^k)| \ge \gamma\|\nabla f(z^k)\|_1 \min_{i=1,\ldots,d}\frac{p_i}{w_i} \\
&\overset{(14)}{\ge} \frac{\gamma}{R_0}\min_{i=1,\ldots,d}\frac{p_i}{w_i}\left(f(z^k) - f(x^*)\right)
\end{aligned}
$$

and

$$\mathbf{E}\left[L_{i_k}(\gamma_i^k)^2 \mid z^k\right] = \gamma^2 \sum_{i=1}^d \frac{L_i p_i}{w_i^2}.$$

Putting it in (62) and taking full expectation from the both sides of obtained inequality we get

$$\mathbf{E}\left[f(z^{k+1}) - f(x^*)\right] \le \left(1 - \frac{\gamma \min\limits_{i=1,\ldots,d}\frac{p_i}{w_i}}{(1-\beta)R_0}\right)\mathbf{E}\left[f(z^k) - f(x^*)\right] + \frac{\gamma^2}{2(1-\beta)^2}\sum_{i=1}^d \frac{L_i p_i}{w_i^2}. \tag{63}$$

Due to our choice of $\gamma \leq \frac{(1-\beta)R_0}{\min\limits_{i=1,\ldots,d} \frac{p_i}{w_i}}$ we have that the factor $\left(1 - \frac{\gamma}{(1-\beta)R_0} \min\limits_{i=1,\ldots,d} \frac{p_i}{w_i}\right)$ is non-negative and, therefore,

$$
\begin{aligned}
\mathbf{E}\left[f(z^k) - f(x^*)\right] &\leq \left(1 - \frac{\gamma}{(1-\beta)R_0} \min_{i=1,\ldots,d} \frac{p_i}{w_i}\right)^k \left(f(z^0) - f(x^*)\right) \\
&\quad + \left(\frac{\gamma^2}{2(1-\beta)^2} \sum_{i=1}^d \frac{L_i p_i}{w_i^2}\right) \sum_{l=0}^{k-1} \left(1 - \frac{\gamma}{(1-\beta)R_0} \min_{i=1,\ldots,d} \frac{p_i}{w_i}\right)^l \\
&\leq \left(1 - \frac{\gamma}{(1-\beta)R_0} \min_{i=1,\ldots,d} \frac{p_i}{w_i}\right)^k \left(f(z^0) - f(x^*)\right) \\
&\quad + \left(\frac{\gamma^2}{2(1-\beta)^2} \sum_{i=1}^d \frac{L_i p_i}{w_i^2}\right) \sum_{l=0}^{\infty} \left(1 - \frac{\gamma}{(1-\beta)R_0} \min_{i=1,\ldots,d} \frac{p_i}{w_i}\right)^l \\
&\leq \left(1 - \frac{\gamma \min\limits_{i=1,\ldots,d} \frac{p_i}{w_i}}{(1-\beta)R_0}\right)^k \left(f(z^0) - f(x^*)\right) + \frac{\gamma R_0}{2(1-\beta) \min\limits_{i=1,\ldots,d} \frac{p_i}{w_i}} \sum_{i=1}^d \frac{L_i p_i}{w_i^2}.
\end{aligned}
$$

Then, putting $\gamma = \frac{\varepsilon(1-\beta) \min\limits_{i=1,\ldots,d} \frac{p_i}{w_i}}{R_0 \sum\limits_{i=1}^d \frac{L_i p_i}{w_i^2}}$ and $k = K$ from (60) in (59) we have

$$
\begin{aligned}
\mathbf{E}[f(z^K)] - f(x^*) &= \left(1 - \frac{\varepsilon \min\limits_{i=1,\ldots,d} \frac{p_i^2}{w_i^2}}{R_0^2 \sum\limits_{i=1}^d \frac{L_i p_i}{w_i^2}}\right)^K \left(f(z^0) - f(x^*)\right) + \frac{\varepsilon}{2} \\
&\leq \exp\left\{-K \cdot \frac{\varepsilon \min\limits_{i=1,\ldots,d} \frac{p_i^2}{w_i^2}}{R_0^2 \sum\limits_{i=1}^d \frac{L_i p_i}{w_i^2}}\right\} \left(f(z^0) - f(x^*)\right) + \frac{\varepsilon}{2} \\
&\overset{(60)}{=} \frac{\varepsilon}{2} + \frac{\varepsilon}{2} = \varepsilon.
\end{aligned}
$$

$\square$

**Theorem E.3** (Decreasing stepsizes). *Let Assumptions 3.2 and 4.1 be satisfied. If we set $\gamma_i^k = \frac{\gamma^k}{w_{i_k}}$ and $\gamma^k = \frac{2}{\alpha k + \theta}$, where $\alpha = \frac{\min\limits_{i=1,\ldots,d} \frac{p_i}{w_i}}{(1-\beta)R_0}$ and $\theta \geq \frac{2}{\alpha}$, then for the iterates of* SMTP_IS *method the following inequality holds:*

$$
\mathbf{E}\left[f(z^k)\right] - f(x^*) \leq \frac{1}{\eta k + 1} \max\left\{f(x^0) - f(x^*), \frac{2}{\alpha\theta(1-\beta)^2} \sum_{i=1}^d \frac{L_i p_i}{w_i^2}\right\}, \tag{64}
$$

*where $\eta \overset{def}{=} \frac{\alpha}{\theta}$. Moreover, if we choose $\gamma^k = \frac{2\alpha}{\alpha^2 k + 2}$ where $\alpha = \frac{\min\limits_{i=1,\ldots,d} \frac{p_i}{w_i}}{(1-\beta)R_0}$ and run* SMTP_IS *for $k = K$ iterations where*

$$
K = \frac{1}{\varepsilon} \cdot \frac{2R_0^2}{\min\limits_{i=1,\ldots,d} \frac{p_i^2}{w_i^2}} \max\left\{(1-\beta)^2(f(x^0) - f(x^*)), \sum_{i=1}^d \frac{L_i p_i}{w_i^2}\right\} - \frac{2(1-\beta)^2 R_0^2}{\min\limits_{i=1,\ldots,d} \frac{p_i^2}{w_i^2}}, \qquad \varepsilon > 0, \tag{65}
$$

*we will get $\mathbf{E}\left[f(z^K)\right] - f(x^*) \leq \varepsilon$.*

*Proof.* In (63) we proved that

$$
\mathbf{E}\left[f(z^{k+1}) - f(x^*)\right] \leq \left(1 - \frac{\gamma \min\limits_{i=1,\ldots,d} \frac{p_i}{w_i}}{(1-\beta)R_0}\right) \mathbf{E}\left[f(z^k) - f(x^*)\right] + \frac{\gamma^2}{2(1-\beta)^2} \sum_{l=1}^d \frac{L_i p_i}{w_i^2}.
$$

Having that, we can apply Lemma C.1 to the sequence $\mathbf{E}\left[f(z^k) - f(x^*)\right]$. The constants for the lemma are: $N = \frac{1}{2(1-\beta)^2}\sum_{l=1}^{d}\frac{L_i p_i}{w_i^2}$, $\alpha = \frac{\min\limits_{i=1,\ldots,d}\frac{p_i}{w_i}}{(1-\beta)R_0}$ and $C = \max\left\{f(x^0) - f(x^*), \frac{2}{\alpha\theta(1-\beta)^2}\sum_{i=1}^{d}\frac{L_i p_i}{w_i^2}\right\}$. Lastly, note that choosing $\gamma^k = \frac{2\alpha}{\alpha^2 k + 2}$ is equivalent to choice $\theta = \frac{2}{\alpha}$. In this case we have $\alpha\theta = 2$ and $C = \max\left\{f(x^0) - f(x^*), \frac{1}{(1-\beta)^2}\sum_{i=1}^{d}\frac{L_i p_i}{w_i^2}\right\}$ and $\eta = \frac{\alpha}{\theta} = \frac{\alpha^2}{2} = \frac{\min\limits_{i=1,\ldots,d}\frac{p_i^2}{w_i^2}}{2(1-\beta)^2 R_0^2}$. Putting these parameters and $K$ from (65) in the (64) we get the result. $\qquad\square$

### E.3 STRONGLY CONVEX CASE

**Theorem E.4** (Solution-dependent stepsizes). *Let Assumptions 3.3 (with $\|\cdot\|_{\mathcal{D}} = \|\cdot\|_1$) and 4.1 be satisfied. If we set $\gamma_i^k = \frac{(1-\beta)\theta_k \min\limits_{i=1,\ldots,d}\frac{p_i}{w_i}}{w_{i_k}\sum\limits_{i=1}^{d}\frac{L_i p_i}{w_i^2}}\sqrt{2\mu(f(z^k) - f(x^*))}$ for some $\theta_k \in (0, 2)$ such that*

$$\theta = \inf_{k\geq 0}\{2\theta_k - \theta_k^2\} \in \left(0, \frac{\sum\limits_{i=1}^{d}\frac{L_i p_i}{w_i^2}}{\mu\min\limits_{i=1,\ldots,d}\frac{p_i^2}{w_i^2}}\right), \text{ then for the iterates of } \texttt{SMTP\_IS} \text{ method the following}$$

*inequality holds:*

$$\mathbf{E}\left[f(z^k)\right] - f(x^*) \leq \left(1 - \frac{\theta\mu\min\limits_{i=1,\ldots,d}\frac{p_i^2}{w_i^2}}{\sum\limits_{i=1}^{d}\frac{L_i p_i}{w_i^2}}\right)^k (f(x^0) - f(x^*)). \tag{66}$$

*If we run* `SMTP_IS` *for $k = K$ iterations where*

$$K = \frac{\sum\limits_{i=1}^{d}\frac{L_i p_i}{w_i^2}}{\theta\mu\min\limits_{i=1,\ldots,d}\frac{p_i^2}{w_i^2}}\ln\left(\frac{f(x^0) - f(x^*)}{\varepsilon}\right), \qquad \varepsilon > 0, \tag{67}$$

*we will get $\mathbf{E}\left[f(z^K)\right] - f(x^*) \leq \varepsilon$.*

*Proof.* Recall that from (52) we have

$$\mathbf{E}\left[f(z^{k+1}) \mid z^k\right] \leq f(z^k) - \frac{1}{1-\beta}\mathbf{E}\left[\gamma_i^k|\nabla_{i_k}f(z^k)| \mid z^k\right] + \frac{1}{2(1-\beta)^2}\mathbf{E}\left[L_{i_k}(\gamma_i^k)^2 \mid z^k\right]. \tag{68}$$

Using our choice $\gamma_i^k = \frac{(1-\beta)\theta_k\min\limits_{i=1,\ldots,d}\frac{p_i}{w_i}}{w_{i_k}\sum\limits_{i=1}^{d}\frac{L_i p_i}{w_i^2}}\sqrt{2\mu(f(z^k) - f(x^*))}$ we derive

$$
\begin{aligned}
\mathbf{E}\left[\gamma_i^k\nabla_{i_k}f(z^k) \mid z^k\right] &= \frac{(1-\beta)\theta_k\min\limits_{i=1,\ldots,d}\frac{p_i}{w_i}}{\sum\limits_{i=1}^{d}\frac{L_i p_i}{w_i^2}}\sqrt{2\mu(f(z^k) - f(x^*))}\sum_{i=1}^{d}\frac{p_i}{w_i}|\nabla_i f(z^k)| \\
&\geq \frac{(1-\beta)\theta_k\left(\min\limits_{i=1,\ldots,d}\frac{p_i}{w_i}\right)^2}{\sum\limits_{i=1}^{d}\frac{L_i p_i}{w_i^2}}\sqrt{2\mu(f(z^k) - f(x^*))}\|\nabla f(z^k)\|_1 \\
&\overset{(20)}{\geq} \frac{2(1-\beta)\theta_k\min\limits_{i=1,\ldots,d}\frac{p_i^2}{w_i^2}}{\sum\limits_{i=1}^{d}\frac{L_i p_i}{w_i^2}}\mu(f(z^k) - f(x^*))
\end{aligned}
$$

and

$$\mathbf{E}\left[L_{i_k}(\gamma_i^k)^2 \mid z^k\right] = \frac{2(1-\beta)^2\theta_k^2 \min\limits_{i=1,\ldots,d} \frac{p_i^2}{w_i^2}}{\left(\sum\limits_{i=1}^d \frac{L_i p_i}{w_i^2}\right)^2}\mu(f(z^k)-f(x^*))\sum_{i=1}^d \frac{L_i p_i}{w_i^2}$$

$$= \frac{2(1-\beta)^2\theta_k^2 \min\limits_{i=1,\ldots,d} \frac{p_i^2}{w_i^2}}{\sum\limits_{i=1}^d \frac{L_i p_i}{w_i^2}}\mu(f(z^k)-f(x^*)).$$

Putting it in (68) and taking full expectation from the both sides of obtained inequality we get

$$\mathbf{E}\left[f(z^{k+1})-f(x^*)\right] \le \left(1-(2\theta-\theta^2)\frac{\mu \min\limits_{i=1,\ldots,d} \frac{p_i^2}{w_i^2}}{\sum\limits_{i=1}^d \frac{L_i p_i}{w_i^2}}\right)\mathbf{E}\left[f(z^k)-f(x^*)\right].$$

Using $\theta = \inf\limits_{k\ge 0}\{2\theta_k-\theta_k^2\} \in \left(0, \frac{\sum\limits_{i=1}^d \frac{L_i p_i}{w_i^2}}{\mu \min\limits_{i=1,\ldots,d} \frac{p_i^2}{w_i^2}}\right)$ we obtain

$$\mathbf{E}\left[f(z^{k+1})-f(x^*)\right] \le \left(1-\frac{\theta\mu \min\limits_{i=1,\ldots,d} \frac{p_i^2}{w_i^2}}{\sum\limits_{i=1}^d \frac{L_i p_i}{w_i^2}}\right)\mathbf{E}\left[f(z^k)-f(x^*)\right]$$

$$\le \left(1-\frac{\theta\mu \min\limits_{i=1,\ldots,d} \frac{p_i^2}{w_i^2}}{\sum\limits_{i=1}^d \frac{L_i p_i}{w_i^2}}\right)^{k+1}\left(f(x^0)-f(x^*)\right).$$

Lasrtly, from (66) we have

$$\mathbf{E}\left[f(z^K)\right]-f(x^*) \le \left(1-\frac{\theta\mu \min\limits_{i=1,\ldots,d} \frac{p_i^2}{w_i^2}}{\sum\limits_{i=1}^d \frac{L_i p_i}{w_i^2}}\right)^K\left(f(x^0)-f(x^*)\right)$$

$$\le \exp\left\{-K\frac{\theta\mu \min\limits_{i=1,\ldots,d} \frac{p_i^2}{w_i^2}}{\sum\limits_{i=1}^d \frac{L_i p_i}{w_i^2}}\right\}\left(f(x^0)-f(x^*)\right)$$

$$\overset{(67)}{\le} \varepsilon.$$

$\square$

The previous result based on the choice of $\gamma^k$ which depends on the $f(z^k)-f(x^*)$ which is often unknown in practice. The next theorem does not have this drawback and makes it possible to obtain the same rate of convergence as in the previous theorem using one extra function evaluation.

**Theorem E.5** (Solution-free stepsizes). *Let Assumptions 3.3 (with $\|\cdot\|_{\mathcal{D}} = \|\cdot\|_2$) and 4.1 be satisfied. If additionally we compute $f(z^k+te_{i_k})$, set $\gamma_i^k = \frac{(1-\beta)|f(z^k+te_{i_k})-f(z^k)|}{L_{i_k}t}$ for $t > 0$, then for the iterates of* `SMTP_IS` *method the following inequality holds:*

$$\mathbf{E}\left[f(z^k)\right]-f(x^*) \le \left(1-\mu \min\limits_{i=1,\ldots,d} \frac{p_i}{L_i}\right)^k\left(f(x^0)-f(x^*)\right) + \frac{t^2}{8\mu \min\limits_{i=1,\ldots,d} \frac{p_i}{L_i}}\sum_{i=1}^d p_i L_i. \quad (69)$$

*Moreover, for any $\varepsilon > 0$ if we set $t$ such that*

$$0 < t \le \sqrt{\frac{4\varepsilon\mu \min\limits_{l=1,\ldots,d} \frac{p_i}{L_i}}{\sum\limits_{i=1}^{d} p_i L_i}}, \tag{70}$$

*and run* `SMTP_IS` *for $k = K$ iterations where*

$$K = \frac{1}{\mu \min\limits_{i=1,\ldots,d} \frac{p_i}{L_i}} \ln\left(\frac{2(f(x^0) - f(x^*))}{\varepsilon}\right), \tag{71}$$

*we will get* $\mathbf{E}\left[f(z^K)\right] - f(x^*) \le \varepsilon$. *Moreover, note that for $p_i = L_i/\sum_i^d L_i$ with $w_i = L_i$, the rate improves to*

$$K = \frac{\sum_{i=1}^{d} L_i}{\mu} \ln\left(\frac{2(f(x^0) - f(x^*))}{\varepsilon}\right). \tag{72}$$

*Proof.* Recall that from (51) we have

$$f(z^{k+1}) \le f(z^k) - \frac{\gamma_i^k}{1-\beta}|\nabla_{i_k} f(z^k)| + \frac{L_{i_k}(\gamma_i^k)^2}{2(1-\beta)^2}.$$

If we minimize the right hand side of the previous inequality as a function of $\gamma_i^k$, we will get that the optimal choice in this sense is $\gamma_{\text{opt}}^k = \frac{(1-\beta)|\nabla_{i_k} f(z^k)|}{L_{i_k}}$. However, this stepsize is impractical for derivative-free optimization, since it requires to know $\nabla_{i_k} f(z^k)$. The natural way to handle this is to approximate directional derivative $\nabla_{i_k} f(z^k)$ by finite difference $\frac{f(z^k + te_{i_k}) - f(z^k)}{t}$ and that is what we do. We choose $\gamma_i^k = \frac{(1-\beta)|f(z^k + te_{i_k}) - f(z^k)|}{L_{i_k} t} = \frac{(1-\beta)|\nabla_{i_k} f(z^k)|}{L_{i_k}} + \frac{(1-\beta)|f(z^k + te_{i_k}) - f(z^k)|}{L_{i_k} t} - \frac{(1-\beta)|\nabla_{i_k} f(z^k)|}{L_{i_k}} \stackrel{\text{def}}{=} \gamma_{\text{opt}}^k + \delta_i^k$. From this we get

$$f(z^{k+1}) \quad \le \quad f(z^k) - \frac{|\nabla_{i_k} f(z^k)|^2}{2L_{i_k}} + \frac{L_{i_k}}{2(1-\beta)^2}(\delta_i^k)^2.$$

Next we estimate $|\delta_i^k|$:

$$\begin{aligned}
|\delta_i^k| &= \frac{(1-\beta)}{L_{i_k} t}\left||f(z^k + te_{i_k}) - f(z^k)| - |\nabla_{i_k} f(z^k)|t\right| \\
&\le \frac{(1-\beta)}{L_{i_k} t}\left|f(z^k + te_{i_k}) - f(z^k) - \nabla_{i_k} f(z^k)t\right| \\
&\stackrel{(26)}{\le} \frac{(1-\beta)}{L_{i_k} t} \cdot \frac{L_{i_k} t^2}{2} = \frac{(1-\beta)t}{2}.
\end{aligned}$$

It implies that

$$\begin{aligned}
f(z^{k+1}) &\le f(z^k) - \frac{|\nabla_{i_k} f(z^k)|^2}{2L_{i_k}} + \frac{L_{i_k}}{2(1-\beta)^2} \cdot \frac{(1-\beta)^2 t^2}{4} \\
&= f(z^k) - \frac{|\nabla_{i_k} f(z^k)|^2}{2L_{i_k}} + \frac{L_{i_k} t^2}{8}
\end{aligned}$$

and after taking expectation $\mathbf{E}\left[\cdot \mid z^k\right]$ conditioned on $z^k$ from the both sides of the obtained inequality we get

$$\mathbf{E}\left[f(z^{k+1}) \mid z^k\right] \le f(z^k) - \frac{1}{2}\mathbf{E}\left[\frac{|\nabla_{i_k} f(z^k)|^2}{L_{i_k}} \mid z^k\right] + \frac{t^2}{8}\mathbf{E}\left[L_{i_k} \mid z^k\right].$$

Note that

$$\begin{aligned}
\mathbf{E}\left[\frac{|\nabla_{i_k} f(z^k)|^2}{L_{i_k}} \mid z^k\right] &= \sum_{i=1}^{d} \frac{p_i}{L_i}|\nabla_i f(z^k)|^2 \\
&\ge \|\nabla f(z^k)\|_2^2 \min_{i=1,\ldots,d} \frac{p_i}{L_i} \\
&\stackrel{(44)}{\ge} 2\mu\left(f(z^k) - f(x^*)\right) \min_{i=1,\ldots,d} \frac{p_i}{L_i},
\end{aligned}$$

since $\|\cdot\|_{\mathcal{D}} = \|\cdot\|_2$, and

$$\mathbf{E}\left[L_{i_k} \mid z^k\right] = \sum_{i=1}^{d} p_i L_i.$$

Putting all together we get

$$\mathbf{E}\left[f(z^{k+1}) \mid z^k\right] \leq f(z^k) - \mu \min_{i=1,\ldots,d} \frac{p_i}{L_i} \left(f(z^k) - f(x^*)\right) + \frac{t^2}{8} \sum_{i=1}^{d} p_i L_i.$$

Taking full expectation from the previous inequality we get

$$\mathbf{E}\left[f(z^{k+1}) - f(x^*)\right] \leq \left(1 - \mu \min_{i=1,\ldots,d} \frac{p_i}{L_i}\right) \mathbf{E}\left[f(z^k) - f(x^*)\right] + \frac{t^2}{8} \sum_{i=1}^{d} p_i L_i.$$

Since $\mu \leq L_i$ for all $i = 1, \ldots, d$ we have

$$
\begin{aligned}
\mathbf{E}\left[f(z^k) - f(x^*)\right] &\leq \left(1 - \mu \min_{i=1,\ldots,d} \frac{p_i}{L_i}\right)^k \left(f(x^0) - f(x^*)\right) \\
&\quad + \left(\frac{t^2}{8} \sum_{i=1}^{d} p_i L_i\right) \sum_{l=0}^{k-1} \left(1 - \mu \min_{i=1,\ldots,d} \frac{p_i}{L_i}\right)^l \\
&\leq \left(1 - \mu \min_{i=1,\ldots,d} \frac{p_i}{L_i}\right)^k \left(f(x^0) - f(x^*)\right) \\
&\quad + \left(\frac{t^2}{8} \sum_{i=1}^{d} p_i L_i\right) \sum_{l=0}^{\infty} \left(1 - \mu \min_{i=1,\ldots,d} \frac{p_i}{L_i}\right)^l \\
&= \left(1 - \mu \min_{i=1,\ldots,d} \frac{p_i}{L_i}\right)^k \left(f(x^0) - f(x^*)\right) + \frac{t^2}{8\mu \min\limits_{i=1,\ldots,d} \frac{p_i}{L_i}} \sum_{i=1}^{d} p_i L_i.
\end{aligned}
$$

Lastly, from (69) we have

$$
\begin{aligned}
\mathbf{E}\left[f(z^K)\right] - f(x^*) &\leq \left(1 - \mu \min_{i=1,\ldots,d} \frac{p_i}{L_i}\right)^K \left(f(x^0) - f(x^*)\right) + \frac{t^2}{8\mu \min\limits_{i=1,\ldots,d} \frac{p_i}{L_i}} \sum_{i=1}^{d} p_i L_i \\
&\overset{(70)}{\leq} \exp\left\{-K\mu \min_{i=1,\ldots,d} \frac{p_i}{L_i}\right\} \left(f(x^0) - f(x^*)\right) + \frac{\varepsilon}{2} \\
&\overset{(71)}{\leq} \frac{\varepsilon}{2} + \frac{\varepsilon}{2} = \varepsilon.
\end{aligned}
$$

$\square$

### E.4 COMPARISON OF SMTP AND SMTP_IS

Here we compare SMTP when $\mathcal{D}$ is normal distribution with zero mean and $\frac{I}{d}$ covariance matrix with SMTP_IS with probabilities $p_i = L_i / \sum_{i=1}^{d} L_i$. We choose such a distribution for SMTP since it shows the best dimension dependence among other distributions considered in Lemma F.1. Note that if $f$ satisfies Assumption 4.1, it is $L$-smooth with $L = \max_{i=1,\ldots,d} L_i$. So, we always have that $\sum_{i=1}^{d} L_i \leq dL$. Table 3 summarizes complexities in this case.

We notice that for SMTP we have $\|\cdot\|_{\mathcal{D}} = \|\cdot\|_2$. That is why one needs to compare SMTP with SMTP_IS accurately. At the first glance, Table 3 says that for non-convex and convex cases we get an extra $d$ factor in the complexity of SMTP_IS when $L_1 = \ldots = L_d = L$. However, it is natural since we use different norms for SMTP and SMTP_IS. In the non-convex case for SMTP we give number of iterations in order to guarantee $\mathbf{E}\left[\|\nabla f(\bar{z}^K)\|_2\right] \leq \varepsilon$ while for SMTP_IS we provide number of iterations in order to guarantee $\mathbf{E}\left[\|\nabla f(\bar{z}^K)\|_1\right] \leq \varepsilon$. From Holder's inequality

| Assumptions on $f$ | SMTP Compleixty | Theorem | Importance Sampling | SMTP_IS Complexity | Theorem |
|---|---|---|---|---|---|
| None | $\frac{\pi r_0 dL}{\varepsilon^2}$ | 3.1 | $p_i = \frac{L_i}{\sum_{i=1}^d L_i}$ | $\frac{2r_0 d \sum_{i=1}^d L_i}{\varepsilon^2}$ | E.1 |
| Convex, $R_0 < \infty$ | $\frac{\pi R_{0,\ell_2}^2 dL}{2\varepsilon} \ln\left(\frac{2r_0}{\varepsilon}\right)$ | 3.2 | $p_i = \frac{L_i}{\sum_{i=1}^d L_i}$ | $\frac{R_{0,\ell_\infty}^2 d \sum_{i=1}^d L_i}{\varepsilon} \ln\left(\frac{2r_0}{\varepsilon}\right)$ | E.2 |
| $\mu$-strongly convex | $\frac{\pi dL}{2\mu} \ln\left(\frac{2r_0}{\varepsilon}\right)$ | 3.5 | $p_i = \frac{L_i}{\sum_{i=1}^d L_i}$ | $\frac{\sum_{i=1}^d L_i}{\mu} \ln\left(\frac{2r_0}{\varepsilon}\right)$ | E.5 |

Table 3: Comparison of SMTP with $\mathcal{D} = N\left(0, \frac{I}{d}\right)$ and SMTP_IS with $p_i = L_i / \sum_{i=1}^d L_i$. Here $r_0 = f(x^0) - f(x^*)$, $R_{0,\ell_2}$ corresponds to the $R_0$ from Assumption C.1 with $\|\cdot\|_\mathcal{D} = \|\cdot\|_2$ and $R_{0,\ell_\infty}$ corresponds to the $R_0$ from Assumption C.1 with $\|\cdot\|_\mathcal{D} = \|\cdot\|_1$.

$\|\cdot\|_1 \leq \sqrt{d}\|\cdot\|_2$ and, therefore, in order to have $\mathbf{E}\left[\|\nabla f(\overline{z}^K)\|_1\right] \leq \varepsilon$ for SMTP we need to ensure that $\mathbf{E}\left[\|\nabla f(\overline{z}^K)\|_2\right] \leq \frac{\varepsilon}{\sqrt{d}}$. That is, to guarantee $\mathbf{E}\left[\|\nabla f(\overline{z}^K)\|_1\right] \leq \varepsilon$ SMTP for aforementioned distribution needs to perform $\frac{\pi r_0 d^2 L}{\varepsilon^2}$ iterations.

Analogously, in the convex case using Cauchy-Schwartz inequality $\|\cdot\|_2 \leq \sqrt{d}\|\cdot\|_\infty$ we have that $R_{0,\ell_2} \leq \sqrt{d} R_{0,\ell_\infty}$. Typically this inequality is tight and if we assume that $R_{0,\ell_\infty} \geq C\frac{R_{0,\ell_2}}{\sqrt{d}}$, we will get that SMTP_IS complexity is $\frac{R_{0,\ell_2}^2 \sum_{i=1}^d L_i}{\varepsilon} \ln\left(\frac{2r_0}{\varepsilon}\right)$ up to constant factor.

That is, in all cases SMTP_IS shows better complexity than SMTP up to some constant factor.

## F AUXILIARY RESULTS

**Lemma F.1** (Lemma 3.4 from Bergou et al. (2019)). *Let $g \in \mathbb{R}^d$.*

1. *If $\mathcal{D}$ is the uniform distribution on the unit sphere in $\mathbb{R}^d$, then*

$$\gamma_\mathcal{D} = 1 \quad and \quad \mathbf{E}_{s \sim \mathcal{D}} |\langle g, s \rangle| \sim \frac{1}{\sqrt{2\pi d}} \|g\|_2. \tag{73}$$

*Hence, $\mathcal{D}$ satisfies Assumption 3.1 with $\gamma_\mathcal{D} = 1$, $\|\cdot\|_\mathcal{D} = \|\cdot\|_2$ and $\mu_\mathcal{D} \sim \frac{1}{\sqrt{2\pi d}}$.*

2. *If $\mathcal{D}$ is the normal distribution with zero mean and identity over $d$ as covariance matrix (i.e. $s \sim N(0, \frac{I}{d})$) then*

$$\gamma_\mathcal{D} = 1 \quad and \quad \mathbf{E}_{s \sim \mathcal{D}} |\langle g, s \rangle| = \frac{\sqrt{2}}{\sqrt{d\pi}} \|g\|_2. \tag{74}$$

*Hence, $\mathcal{D}$ satisfies Assumption 3.1 with $\gamma_\mathcal{D} = 1$, $\|\cdot\|_\mathcal{D} = \|\cdot\|_2$ and $\mu_\mathcal{D} = \frac{\sqrt{2}}{\sqrt{d\pi}}$.*

3. *If $\mathcal{D}$ is the uniform distribution on $\{e_1, \ldots, e_d\}$, then*

$$\gamma_\mathcal{D} = 1 \quad and \quad \mathbf{E}_{s \sim \mathcal{D}} |\langle g, s \rangle| = \frac{1}{d} \|g\|_1. \tag{75}$$

*Hence, $\mathcal{D}$ satisfies Assumption 3.1 with $\gamma_\mathcal{D} = 1$, $\|\cdot\|_\mathcal{D} = \|\cdot\|_1$ and $\mu_\mathcal{D} = \frac{1}{d}$.*

4. *If $\mathcal{D}$ is an arbitrary distribution on $\{e_1, \ldots, e_d\}$ given by $\mathbf{P}\{s = e_i\} = p_i > 0$, then*

$$\gamma_\mathcal{D} = 1 \quad and \quad \mathbf{E}_{s \sim \mathcal{D}} |\langle g, s \rangle| = \|g\|_\mathcal{D} \overset{def}{=} \sum_{i=1}^d p_i |g_i|. \tag{76}$$

*Hence, $\mathcal{D}$ satisfies Assumption 3.1 with $\gamma_\mathcal{D} = 1$ and $\mu_\mathcal{D} = 1$.*

5. *If $\mathcal{D}$ is a distribution on $D = \{u_1, \ldots, u_d\}$ where $u_1, \ldots, u_d$ form an orthonormal basis of $\mathbb{R}^d$ and $\mathbf{P}\{s = d_i\} = p_i$, then*

$$\gamma_\mathcal{D} = 1 \quad and \quad \mathbf{E}_{s \sim \mathcal{D}} |\langle g, s \rangle| = \|g\|_\mathcal{D} \overset{def}{=} \sum_{i=1}^d p_i |g_i|. \tag{77}$$

*Hence, $\mathcal{D}$ satisfies Assumption 3.1 with $\gamma_\mathcal{D} = 1$ and $\mu_\mathcal{D} = 1$.*

