# OpenReview forum: "A Stochastic Derivative Free Optimization Method with Momentum"
_ICLR.cc/2020/Conference — Accept (Poster)_

### Official Review · AnonReviewer1 · 2019-10-22
**Official Blind Review #1**

**Rating:** 6

**Review:**

The paper proposes a stochastic derivative free optimization algorithm. The contribution is two-fold: first, the paper introduces the heavy ball momentum into the STP framework; second, the paper fulfills both the importance sampling and heavy ball momentum in the STP framework. For both methods, the paper provides the convergence guarantees and rates. The experiments on reinforcement learning data-sets, as compared with the original STP, shows improvement.

The idea seems straightforward --- just combining a classical momentum strategy with the an existing derivative free optimization framework. But the author claim that they are the first to exploit this strategy. The analysis part, for strongly convex, convex and nonconvex problems, however, is solid to me. I am not the expert in this direction. Here are a few questions, from the answers of which I want to learn more about the meaning of this work.

(1) As compared with other derivative free optimization algorithm, such as Bayesian optimization/genetic algorithms/simulated annealing, what is the advantage of the proposed method and also the STP framework?
(2) The experiments seem weak to me. Why does the paper only compare with STP? Are there any other baselines, such as stochastic two points, BO and GA? Is it possible to conduct evaluation on other applications? For example, some general optimization tasks but not allowing gradient calculation?

**Experience Assessment:**

I do not know much about this area.

**Review Assessment: Checking Correctness Of Derivations And Theory:**

I did not assess the derivations or theory.

**Review Assessment: Checking Correctness Of Experiments:**

I assessed the sensibility of the experiments.

**Review Assessment: Thoroughness In Paper Reading:**

I read the paper at least twice and used my best judgement in assessing the paper.

---

> ### Author Response · Authors · 2019-11-15
> **Response to R1**
>
> We thank R1 for their positive feedback and acknowledgment to our contributions. All comments have been addressed in blue in the revised version. Follows our response.
>
> $\bullet$ "(1) As compared with other derivative free optimization algorithm, such as Bayesian optimization/genetic algorithms/simulated annealing, what is the advantage of the proposed method and also the STP framework?"
>
>
> As we mentioned in our response to R3 there are many DFO methods and comparing against all of them is beyond the reach of our paper. Advantage of {\tt STP} framework in general is in its simplicity and generality (see the paragraph in the introduction devoted to {\tt STP}).
>
> $\bullet$ "(2) The experiments seem weak to me. Why does the paper only compare with STP? Are there any other baselines, such as stochastic two points, BO and GA? Is it possible to conduct evaluation on other applications? For example, some general optimization tasks but not allowing gradient calculation?"
>
>
> In the experiments section, we compare against several algorithms on the MuJoCo task and not only against STP. For instance, in Table 2, we report comparisons against 2 policy gradient methods (NG-lin) and (TRPO-nn). Moreover, we compare against ARS(V1-t) and ARS(V2-t) which to the best of our knowledge achieve the state-of-art results on the respective environments.

---

### Official Review · AnonReviewer3 · 2019-10-26
**Official Blind Review #3**

**Rating:** 6

**Review:**

The authors extend recent the recent stochastic three-point (STP) method to allow for Polyak-style momentum, as well as momentum with importance sampling. They provide a range of analysis that mostly extends existing STP results to the STP+momentum case. Most of these results are similar in spirit to stochastic gradient or subgradient results, as in the methods converge up to a ball around the solution, with radius depending on step-size, so you can get an epsilon solution by choosing a suitably small stepsize. The analysis covers non-convex cases (bounds are on the norm of the gradient) and the importance sampling case as well.

Overall, I think this is a strong paper, and a very interesting topic, and hence I support a "Weak accept". The numerical results look good, as the new method outperforms most of the compared methods, at least for the easier problems (SMTP beats competitor ARS in 3/5 trials; both methods are generally very similar, though SMTP does better on the easy Swimmer problem; the SMTP_IS results are more complicated).  The analysis is mostly good (non-trivial), and shows a broad understanding.

That said, I have some concerns.

(1) The analysis is nice in that it shows the methods work, but doesn't demonstrate benefit of their method over other methods

(2) Given that there are no results showing this method has better worst-case rates than other methods, we rely on experiments to see the actual benefit. In this case, more experiments is always better.

(3) I am quite skeptical of the importance sampling scheme. It's nice to include it, but I don't think it strengthens the paper too much. Empirically, the performance seems to help sometimes but not other times. Finding the individual Lipschitz constants seems tricky; this paper re-uses a scheme that iterates for a while, fits a function, and uses that to estimate the constants (it wasn't clear if this pre-processing was counted in the iteration count for experimental results). It's not clear how well that works to get an accurate estimate. Furthermore, to exploit the importance sampling, the directions must be sampled from a pre-determined basis, which seems restrictive.  This criticism is not just of the current paper but of other papers that use this approach.


-- The manuscript needs more proof reading, as there are mistakes in most paragraphs. There are a lot of problems with missing articles.  Phrases like "results for STP are shorts and clear" ("shorts" --> "short"), "which updates rule" [?? which-->with?? ], "hints [at] the update rule", "is far more superior" [-->"is far superior", since you can't be more superior], etc.

-- There is a confusion over how to use \cite, \citet and \citep in latex. Given the bibtex citation style, this makes it very hard to read in places

-- Literature review seems good and pretty thorough (mentions most key references through 2015, and a good selection of references since then).

-- Assumption 3.1 part 2 is stated in a funny way (it says, "there is a constant mu_D and a norm || ||_D such that ...").  You are free to choose the norm, and then find the constant (since all norms in finite dimensions are equivalent). That way, you can choose the norm that gives the tightest inequality.  I think you are aware of this, and it's just a wording issue.

-- Theorem 3.5 (Thm D.2) requires the mu_D^2 to be less than the condition number, which is weird. The easier the problem is, the tighter your assumptions are.  I suspect that this is because you use an inequality somewhere that simplifies things by bounding a term by the condition number. But as stated, this is a weak theorem.  It is also confusing because you have a mu_D which is not the strong convexity constant, but the actual strong convexity constant (mu) *does* depend on the norm D (cf eq 19; and this must be so, otherwise you can cheat and then the value of mu_D is meaningless). So both mu's are functions of the norm D. However, the Lipschitz constant L is *not* a function of D. So notation is confusing and makes interpreting the results harder.

-- sentences like "We achieve the state-of-the-art performance compared to *all* DFO based and policy gradient methods" are in appropriate (*italics* are mine). You mean to say that on the few examples you ran, based on a few DFO and policy gradient methods you tested, that the best of your two methods was better than the competitor methods on 4/5 problems.

-- I think a common-sense algorithm to compare to would be gradient descent (or heavy-ball) using finite differences to estimate the gradient.  In small dimensions this isn't such a bad idea.  I don't actually know what the dimensions of your test problems are (I looked in section 5 but didn't see it mentioned, other than reference to Ant-v1 and Humanoid-v1 being "high dimensional"; I think this is extremely relevant information. In small dimensions, traditional DFO and Bayesian optimization methods are competitive).

-- p 26/27, "Causchy-Schwartz" is spelled wrong, and usually this is called "Holder's inequality" when it's not the Euclidean norm.

-- Table 3, there is no space between the caption and the main text, so it's confusing

-- Eq (76) in appendix, the sum should go to d not n.

-- I think s^k may need to be independent of z^k for their tower property thing to work, otherwise it's not clear what's happening with the inequality prior to overset (30) on that last line of pg17. For example, if s^k were z^k measurable then that whole thing in the inner expectation would be a constant. This isn't a problem, it's fairly natural to assume that s^k is independent of z^k, I just didn't see the assumption anywhere.

-- Overall, comparing importance sampling results is hard, due to the different norms (this is mentioned in the paper, and there are inequalities between norms, but it's still hard to get a good result that shows importance sampling has better worst-case rates).


== AFTER READING REBUTTAL ==
I read the authors' response, and I am still slightly positive about the paper, though my major points were not addressed, but mostly deflected, e.g., referring to other papers that claim to show benefits of importance sampling. I think all the reviewers were curious how Bayesian optimization (BO) would perform. We understand numerical experiments are time consuming, but it's disappointing that you're not curious yourself whether your method outperforms BO. Your basic deflection seems to be that BO doesn't have provable guarantees, so because you do have guarantees, you don't need to compare with it.  Having one of the fastest methods among all methods with provable guarantees, but not necessarily the fastest method in general, sounds like a consolation prize to me.

The revision did not address some of the minor issues I mentioned, such as confusing "cite"/"cited"/"citep" issues in latex (which makes it hard to read).  My comment about Holder's inequality (vs Cauchy-Schwarz) applies not just to the $\|\cdot\|_1 \le \sqrt{d}\|\cdot\|_2$ bound, but also to the  $\|\cdot\|_2 \le \sqrt{d}\|\cdot\|_\infty$ bound in the next paragraph.

But despite a few items I'm being cranky about, I think it's still a solid paper, and it's still a weak accept.

**Experience Assessment:**

I have published one or two papers in this area.

**Review Assessment: Checking Correctness Of Derivations And Theory:**

I assessed the sensibility of the derivations and theory.

**Review Assessment: Checking Correctness Of Experiments:**

I assessed the sensibility of the experiments.

**Review Assessment: Thoroughness In Paper Reading:**

I read the paper thoroughly.

---

> ### Author Response · Authors · 2019-11-15
> **Response to R3**
>
> We thank R3 for their constructive thorough detailed review. Note that all typos and minor comments have been addressed in blue in the revised version. Follows our response.
>
> $\bullet$(1) "The analysis is nice in that it shows the methods work, but doesn't demonstrate benefit of their method over other methods"
>
> Indeed, we do not confirm theoretically that ${\tt SMTP}$ outperforms ${\tt STP}$. However, we mentioned that for the general case of objectives it is still an open question whether Heavy ball method outperforms Gradient Descent theoretically. Since ${\tt STP}$ can be considered as zeroth-order variant of Gradient Descent and ${\tt SMTP}$ as zeroth-order variant of Heavy ball method, it is natural to have no benefits of ${\tt SMTP}$ over ${\tt STP}$, at least theoretically. But still it was needed to show that ${\tt SMTP}$ is not worse in terms of theoretical convergence rates than ${\tt STP}$ in order to have some guarantees for the new method and verify that it relates to ${\tt STP}$ in the same way as Heavy ball methods relate to Gradient Descent.
>
> $\bullet$ (2) "Given that there are no results showing this method has better worst-case rates than other methods, we rely on experiments to see the actual benefit. In this case, more experiments is always better."
>
> An advantage of the proposed algorithm is that it indeed enjoys a provable convergence rate. None of the competitors, except ${\tt STP}$ and ${\tt STP{\_}IS}$, enjoy any theoretical results for the rate of convergence but they have been well celebrated for their excellent performance on the MuJoCo environments. In this work, the proposed algorithm enjoys both the theoretical rates and practicality that outperforms several competition. The complexity of the experiments is comparable and match previous art (Rajeswaran et al. conduct experiments on 6 environments while Schulman et al. on 7). We conduct experiments on 5 environments but provide convergence guarantees.
>
>
>
>
> $\bullet$ "(3) I am quite skeptical of the importance sampling scheme. It's nice to include it, but I don't think it strengthens the paper too much. Empirically, the performance seems to help sometimes but not other times. Finding ... "
>
> It has been observed by Bibi et. al. that importance sampling with STP vastly improves upon uniform sampling. In this work and upon merging both momentum and importance sampling such conclusion is indeed far less obvious as pointed out by R3. We provide such analysis for the sack of completion and leave most of the details to the supplementary material. The preprocessing of retraining the smooth function is very negligible compared (order of milliseconds) to the reward function evaluation (simulator run) that is of order of seconds.
>
>
> $\bullet$"Theorem 3.5 (Thm D.2) requires the $\mu_D^2$ to be less than the condition number, which is weird. The easier the problem is, the tighter your assumptions are.  I suspect that this is because you use an inequality somewhere that simplifies things by bounding a term by the condition number. But as ..."
>
> Actually, it is not so weird assumption. Please, see the Lemma~F.1. It covers 5 examples of distributions $\mathcal{D}$ that fit Asumption~3.1. Note that for the first two examples $\mu_{\mathcal{D}}$ is less than $1$ and $\|\cdot\|_{\mathcal{D}} = \|\cdot\|_2 = \|\cdot\|_{\mathcal{D}}^*$, so, we always have $\mu_{\mathcal{D}}^2 \le 1 \le \frac{L}{\mu}$ for these cases. For the third case we have $\|\cdot\|_{\mathcal{D}} = \|\cdot\|_1$, $\|\cdot\|_{\mathcal{D}}^* = \|\cdot\|_\infty$, $\mu_{\mathcal{D}} = \frac{1}{d}$ and due to classical relation $\|x\|_2 \le \sqrt{d}\|x\|_\infty$ we get that if the function $f$ is $\mu$-strongly convex in $\ell_\infty$-norm then it is $\hat{\mu}$-strongly convex in $\ell_2$-norm with $\hat{\mu} \ge \frac{\mu}{d}$. Using this we get $\frac{L}{\mu} \ge \frac{L}{\hat\mu d} \ge \frac{1}{d} \ge \frac{1}{d^2} = \mu_{\mathcal{D}}^2$ since $L \ge \hat\mu$.
>
> Thank you so much for your comment, it helped us to find a small typo in Assumption~3.1, see the revised version.
>
>
> $\bullet$ "sentences like "We achieve the state-of-the-art performance compared to *all* DFO based and policy gradient methods" are in appropriate (*italics* are mine). You mean ..."
>
> The statement has been tuned down the final version.

---

### Official Review · AnonReviewer2 · 2019-10-28
**Official Blind Review #2**

**Rating:** 6

**Review:**

This paper studies the so called problem of derivative-free optimization, which is relevant for cases when the evaluation function is continuous but access to gradients is not possible. The paper improves on top of the stochastic three points method (STP), an existing work (published in arXiv), by proposing adding momentum (SMTP). The intuition behind both STP and SMTP is rather straighforward: you sample a random direction s, then given your current position x you check x+as and x-as. You then move to the best position from (x, x+as, x_as). In a way, this is like computing the numerical derivatives (instead of the gradient) given a random location and its mirror, and then applying gradient descent given the best numerical derivative. However, take this analogy with a large grain of salt, as there are many differences with GD. The proposed algorithm adds momentum and importance sampling. Momentum helps speed up convergence, as the paper shows for non-convex, convex and strongly convex functions. All three cases are individually examined and bounds are derived regarding the speed of convergence. For the non-convex case the speed of convergence is 1/\sqrt{K}, K being the number of iterations. For the convex case it is 1/K. For the strongly convex case the (unrealistic) assumption of knowing the optimal value is removed while maintaining the same speed of convergence.  Importance sampling helps computing the derivatives focusing on those coordinate dimensions that are more critical to the objective function f(x), improving the speed of convergence further. The importance sampling is proportional to the coordinate-wise Lipschitz constants, assuming that the objective function is coordinate-wise Lipschitz smooth. The methods are validated on five different cases of MuJoCo. Results seem good when compared to the STP ones. Compared to policy gradient methods, the results seem much better.

Strengths:
+ The paper presents a small but interesting and well-motivated addition to the original algorithm STP. I particularly liked how straightforward the final algorithm is: applying momentum and sampling according to the Lipschitz constants.

+ At least at a first glance the results look good. Compared to STP in figure 1 there is a clear improvement not only in the final optimum but also in the speed of attaining the said optimum.

+ I liked a lot the presentation and clarity of writing. While quite mathematically dense, it was easy to follow the big story and understand that underlying points.

Weaknesses:
+ While interesting and useful, I am not completely convinced whether the added novelty over (Bergou et al, 2019) is significant enough. At the end of the day, the final algorithm is the conglomeration of two existing algorithms, that is STP and momentum. STP is very similar to the final algorithm, after all it is the basis for it. The authors argue that it is not trivial to select the next points under the momentum term. To this end, they propose to rely on yet another existing approach, that is the virtual iterates analysis from (Yang et al. 2016). However, it is not clear why these points are "optimal", what is so "non-trivial" about selecting them? This is basically skimmed over in two lines.

+ In the strongly convex case one assumption (knowing the f(x*) ) is replaced with another assumption, that all points lie on a hypersphere (|s|_2=1). I suppose this would assume a spherical normalization of the input space. While this is not an unrealistic assumption, it does place a constraint which could be problematic in the case of high dimensions for s? In that case the high dimensionality would render distances rather unreliable and in turn could hurt convergence? This is also perhaps the reason that only the MuJoCo enviroments were tested? In general, I would say that the strongly convex case was discussed less clearly and it is not exactly clear the final result. In the end, eq (25) does contain f(x*), whereas in the convex case K does not (K \approx 2 R_0^2 L γ_D/(εμ_D^2).

+ Some statements are unclear.
  ++ In p. 2 some symbols are not explained, e.g., ε. While it is quite clear for peopled versed in the field, in my opinion it is bad practice to leave notation not explained.
  ++ In assumption 3.1 seems rather trivial? Wouldn't γ_D by definition be always positive, since is the expectation of a squared norm (always positive)? Does this need to be an assumption?
  ++ Between eq. (11) and (12) there is reference to (35)? What is (35)?
  ++ It is not clear in practice how the importance sampling is performed. In Algorithm 2 the probabilities p_i are defined as function inputs and then never updated. Is that true? If yes, how is p_i decided in the first place? What is the connection to the Lipschitz constants L_i?

+ A highly relevant field appears to be Bayesian Optimization, where also one cannot compute gradients and must optimize a black-box function. Some relevant recent works are [1] and [2] for continuous and discrete inputs. It would be interesting to discuss what are the distinct differences with bayesian optimization methods in [1] and [2].

+ I would say that the paper is rather on the light side regarding experiments. Only MuJoCo is used as an experimental setup. It would be nice to also report results on synthetic experiments with known functions to better understand the limitations of the algorithm. Synthetic and realistic setups can be found in [1] and [2].

What is more, the experimental choices are not entirely clear. What is the "predefined reward threshold" and why was that chosen? For instance, the leaderboard for "Swimmer" is in: https://www.endtoend.ai/envs/gym/mujoco/swimmer/. How does the proposed algorithm fair compared to these works? Also, *maybe* it would be interesting to compare even against [1] or [2] (I guess [2] is harder as it is for discete inputs), assuming that a relatively low number of iterations is performed.

[1] BOCK: Bayesian Optimization with Cylindrical Kernels, C. Oh, E. Gavves, M. Welling, ICML 2018
[2] BOCS: Bayesian Optimization of Combinatorial Structures, R. Baptista, M. Poloczek, ICML 2018


**Experience Assessment:**

I have published one or two papers in this area.

**Review Assessment: Checking Correctness Of Derivations And Theory:**

I assessed the sensibility of the derivations and theory.

**Review Assessment: Checking Correctness Of Experiments:**

I carefully checked the experiments.

**Review Assessment: Thoroughness In Paper Reading:**

I read the paper at least twice and used my best judgement in assessing the paper.

---

> ### Author Response · Authors · 2019-11-15
> **Response (1/2) to R2**
>
> We thank R2 for their constructive thorough detailed review. Follows our response. All edits are marked in blue in the revised version.
>
>
> $\bullet$ "While interesting and useful, I am not completely convinced whether the added novelty over (Bergou et al, 2019) is significant enough. At the end of the day, the final algorithm is the conglomeration of two existing algorithms, that is STP and momentum. STP is very similar to the ..."
>
>
> Note that we claim nothing about optimality of our approach in the paper. However, we agree that investigating different approach beyond stochastic three points is an interesting direction of future work. By non-triviality of our approach we mean that ${\tt SMTP}$ is not a straightforward ${\tt STP}$-like modification of Polyak's method: instead of classical form of the Heavy ball method we use equivalent form from (Yang et al. 2016) and choose next iterate $x^{k+1}$ not as argminimum of $x^k, x_{+}^{k+1}, x_{-}^{k+1}$, but use virtual iterates instead.
>
>
>
> $\bullet$ "In the strongly convex case one assumption (knowing the $f(x*)$ ) is replaced with another assumption, that all points lie on a hypersphere ($\|s\|_2=1$). I suppose this would assume a spherical normalization of the input space. While this is not an unrealistic assumption, it does ..."
>
>
>
> We do not think that Assumption~3.4 from our paper is unrealistic or restrictive. Indeed, in the case of high-dimensions it can cause additional problems connected with normalization. However, in the high-dimensional case the method itself works slow which is the common ``disease of DFO methods and additional normalization for this case does not change the situation dramatically. According your comment about our analysis in the strongly convex case itself -- yes, due to space limitations we do not introduce some classical fact about $\mu$-strongly convex and $L$-smooth problems. For example, classical relations $\frac{\mu}{2}\|x^0 - x^*\|_{{\mathcal{D}}}^2 \le f(x^0) - f(x^*)$ and $f(x^0) - f(x^*) \le \frac{L}{2}\|x^0 - x^*\|_2^2$ are the answer for your question: in this case $R_0^2$ and $f(x^0) - f(x^*)$ are equivalent up to some constants and it does not play a big role since $f(x^0) - f(x^*)$ appears in (25) under the logarithm.
>
>
> $\bullet$ Defining $\epsilon$
>
> We added a definition for $\epsilon$ the first time it is introduced (Theorem 3.2).
>
> $\bullet$ Assumption 3.1
>
> We have restated Assumption 3.1 to address R2's comments.
>
> $\bullet$ "Between eq. (11) and (12) there is reference to (35)? What is (35)?"
>
> Equation (35) is identical to Eq (11) but was rederived in the supplementary material. We have corrected the reference to Equation (11).
>
>
>
> $\bullet$ "It is not clear in practice how the importance sampling is performed. In Algorithm 2 the probabilities $p_i$ are defined as function inputs and ..."
>
> That is correct. The probabilities $p_i$ are computed once before the algorithm and never updated and they are a function of $L_i$. Table 1 summarizes the choice of $p_i$. Note that in the supplementary material, we derive the rates for nonconvex (Theorem E.1), convex (Theorems E.2 and E.3) and strongly convex (Theorems E.4 and E.5) problems as a function of arbitrary sampling probabilities $p_i$. We propose the importance sampling strategy (proportional to $L_i$) as depicted in Table 1, to show that this strategy enjoys better worst complexity rate than uniform sampling. For non-convex problems of the MuJoCo experiments, $L_i$s are not known apriori for the reward function. Thus we follow, section E of Bibi et. al. and approximate the objective function with a smooth parametric family of neural networks where we can estimate the smoothness constants $L_i$.
>
>
>
> $\bullet$ "A highly relevant field appears to be Bayesian Optimization, where also one cannot compute gradients and must optimize a black-box function. Some relevant recent ... "
>
>
>
> To the best of our knowledge, Bayesian optimization is about global optimization of black box functions where there are no necessary assumptions regarding smoothness. The flavor of our work is slightly different where we have convergence rate while we are not aware of any for Bayesian optimization.
>
> The most related methods to us from the literature are STP and the methods mentioned in the STP paper (deterministic direct search, random gradient free method, direct search based on random directions). We had a comparison of STP with them on the STP paper. STP was outperforming all compared methods and since $STP_{\text{momentum}}$ is outperforming STP, this demonstrates the superiority of our proposed algorithm compared to all methods of the same class.

---

> > ### Author Response · Authors · 2019-11-15
> > **Response (2/2) to R2**
> >
> >
> >
> >
> > $\bullet$ "I would say that the paper is rather on the light side regarding experiments. Only MuJoCo is used as an experimental setup. It would be nice to also ... "
> >
> >
> > We believe that the complexity of the experiments either matches or exceeds the experiments reported in the stochastic optimization literature that provide provably convergence algorithms with rates. That is to say, we believe that the real experiments on real data (continuous control MuJoCo experiments) are the major factor here. We believe this speak about the quality of our work as a whole. We will take the reviewers comment into consideration in the final version of the paper.
> >
> >
> > $\bullet$ "What is more, the experimental choices are not entirely clear. What is the "predefined reward  ..."
> >
> > The predefined reward thresholds were previously proposed in the literature of continuous control community as such that MuJoCo agents are considered to have successfully completed the task if they achieve these predefined thresholds for the reward function or higher. See ARS paper for reference.

---

### Decision · Program_Chairs · 2019-12-19

**Decision:**

Accept (Poster)

**Comment:**

A new method for derivative free optimization including momentum and importance sampling is proposed.

All reviewers agreed that the paper deserves acceptance.

Acceptance is recommended.